# SARS-CoV-2 spike glycoprotein vaccine candidate NVX-CoV2373 immunogenicity in baboons and protection in mice

Jing-Hui Tian[1,7], Nita Patel[1,7], Robert Haupt[2,7], Haixia Zhou[1], Stuart Weston [2], Holly Hammond[2], James Logue [2], Alyse D. Portnoff[1], James Norton[1], Mimi Guebre-Xabier[1], Bin Zhou[1], Kelsey Jacobson[1], Sonia Maciejewski [1], Rafia Khatoon[1], Malgorzata Wisniewska[1], Will Moffitt[1], Stefanie Kluepfel-Stahl[1], Betty Ekechukwu[1], James Papin [3], Sarathi Boddapati[4], C. Jason Wong[4], Pedro A. Piedra[5], Matthew B. Frieman [2], Michael J. Massare[1], Louis Fries[1], Karin Lövgren Bengtsson[6], Linda Stertman[6], Larry Ellingsworth[1], Gregory Glenn[1] & Gale Smith [1✉]

The COVID-19 pandemic continues to spread throughout the world with an urgent need for a safe and protective vaccine to effectuate herd protection and control the spread of SARS-CoV-2. Here, we report the development of a SARS-CoV-2 subunit vaccine (NVX-CoV2373) from the full-length spike (S) protein that is stable in the prefusion conformation. NVX-CoV2373 S form 27.2-nm nanoparticles that are thermostable and bind with high affinity to the human angiotensin-converting enzyme 2 (hACE2) receptor. In mice, low-dose NVX-CoV2373 with saponin-based Matrix-M adjuvant elicit high titer anti-S IgG that blocks hACE2 receptor binding, neutralize virus, and protects against SARS-CoV-2 challenge with no evidence of vaccine-associated enhanced respiratory disease. NVX-CoV2373 also elicits multifunctional CD4$^+$ and CD8$^+$ T cells, CD4$^+$ follicular helper T cells (Tfh), and antigen-specific germinal center (GC) B cells in the spleen. In baboons, low-dose levels of NVX-CoV2373 with Matrix-M was also highly immunogenic and elicited high titer anti-S antibodies and functional antibodies that block S-protein binding to hACE2 and neutralize virus infection and antigen-specific T cells. These results support the ongoing phase 1/2 clinical evaluation of the safety and immunogenicity of NVX-CoV2373 with Matrix-M (NCT04368988).

[1] Novavax, Inc. 21 Firstfield Road, Gaithersburg, MD 20878, USA. [2] University of Maryland, School of Medicine 685 West Baltimore St, Baltimore, MD 21201, USA. [3] Department of Pathology, Division of Comparative Medicine, University of Oklahoma, Health Sciences Center 940 Stanton L. Young, BMS 203, Oklahoma City, OK 73104, USA. [4] Catalent Cell & Gene Therapy 20 Firstfield Road, Gaithersburg, MD 20874, USA. [5] Department of Molecular Virology and Microbiology, and Pediatrics, Baylor College of Medicine Houston, TX, USA. [6] Novavax AB Kungsgatan 109, SE-753 18 Uppsala, Sweden. [7] These authors contributed equally: Jing-Hui Tian, Nita Patel, Robert Haupt. ✉email: GSmith@Novavax.com

Rapid global transmission of SARS-CoV-2 has followed the initial outbreak in Wuhan, Hubei Province, China, first reported in December 2019. The World Health Organization's (WHO) 25 October 2020 COVID-19 Situation Report reports 44 million confirmed cases worldwide and over 1 million deaths[1,2]. Current estimates suggest a substantial asymptomatic incubation period during which transmission occurs, and a basic reproduction number ($R0$) of 2.23–2.51 (ref. [3]), greater than any twentieth or twenty-first century pandemic influenza virus. The urgent need for a safe, effective, stable, globally deployable, preventative vaccine has led to an unprecedented collaboration between vaccine developers, manufacturers, and distributors in concert with government and academic programs[4].

The SARS-CoV-2 spike (S) glycoprotein is a major component of the virus envelope, essential for receptor binding, fusion, virus entry, and a target of host immune defense[5–9]. The SARS-CoV-2 S glycoprotein is a class I fusion protein produced as a large 1273 amino acid inactive precursor (S0). Unique to SARS-CoV-2 is the insertion of a polybasic RRAR furin-like cleavage motif in the S1/S2 cleavage site[10]. Proteolytic cleavage of the S protein generates the S2 stalk that is conserved across human coronaviruses and the less conserved S1 cap[11]. The N-terminal domain (NTD) and the receptor-binding domain (RBD) are located in the S1 subunit. The fusion peptide (FP), two heptad repeats (HR1 and HR2), central helix (CH), transmembrane (TM) domain, and cytoplasmic tail (CT) are located in the S2 subunit. Three S1/S2 protomers non-covalently associate to form the functional S-trimer. Like other fusion proteins, the SARS-CoV-2 S-trimer is metastable and undergoes significant structural rearrangement from a prefusion conformation to a thermostable postfusion conformation upon S-protein receptor binding and proteolytic cleavage, either at the plasma membrane or following endocytosis[12]. Rearrangement exposes the hydrophobic FP allowing insertion into the host cell membrane, facilitating virus/host cell membrane alignment, fusion, and virus entry[13–16].

We have developed a SARS-CoV-2 S subunit vaccine (NVX-CoV2373) constructed from the full-length S-protein and produced in the established Sf9 insect cell expression system. Here, we describe a stable prefusion S-protein structure generated by mutating the furin cleavage site to be resistant to cleavage and utilization of two proline substitutions at the apex of the CH[11]. Here we show that administering NVX-CoV2373 with Matrix-M adjuvant in a nonhuman primate and mice models induces a B- and T-cell responses, hACE2 (human angiotensin-converting enzyme 2)-receptor-blocking antibodies, and SARS-CoV-2-neutralizing antibodies. In mice, the vaccine elicits protection against SARS-CoV-2 infection with no evidence of vaccine-associated enhanced respiratory disease (VAERD). These results support the clinical development of the NVX-CoV2373 vaccine for prevention of COVID-19 (NCT04368988).

## Results

**SARS-CoV-2 S glycoproteins.** The SARS-CoV-2 S-gene (MN908947.3, nucleotides 21563–25384) encoding the full-length 1273 amino acid spike protein was used as a backbone to produce spike protein variants. The BV2365 single mutant was generated by mutating the putative furin cleavage site 682-RRAR-685 to 682-QQAQ-685, and the NVX-CoV2373 double mutant was generated with 682-QQAQ-685 and two proline substitutions at residues K986P and V987P (Fig. 1a). Synthetic full-length wild-type (WT), the single mutant BV2365, and double mutant NVX-CoV2373 genes were codon optimized for insect cells and cloned into recombinant baculovirus for expression in Sf9 cells.

**Biophysical characterization and stability.** Purified SARS-CoV-2 WT, BV2365, and NVX-CoV2373 S-proteins, when reduced, migrated with an apparent molecular weight of 180 kDa (Fig. 1b). Expression of WT spike in Sf9 cells results in a mixture of cleaved and uncleaved glycoprotein. This is evident by western blot analysis with an anti-S2 specific antibody which shows both cleaved and uncleaved spike protein in partially purified membrane extracts. Dynamic light scattering (DLS) showed the WT S-protein had a Z-average particle diameter of 69.53 nm compared to a twofold smaller particle size of BV2365 (33.4 nm) and NVX-CoV2373 (27.2 nm). The polydispersity index (PDI) indicated that BV2365 and NXV-CoV2373 particles were generally uniform in size, shape, and mass (PDI = 0.25–0.29) compared to the WT spike protein (PDI = 0.46) (Table 1).

The thermal stability of the S-trimers was determined by differential scanning calorimetry (DSC). The thermal transition temperature of the WT S-spike ($T_{max} = 58.6\,°C$) was similar to BV2365 and NXV-CoV2373 with a $T_{max} = 61.3\,°C$ and $60.4\,°C$, respectively (Table 1). Of greater significance was the 3–5-fold increased enthalpy of transition required to unfold the BV2365 and NXV-CoV2373 variants ($\Delta H$cal = 466 and 732 kJ mol$^{-1}$, respectively) compared to the lower enthalpy required to unfold the WT spike protein ($\Delta H$cal = 153 kJ mol$^{-1}$). These results are consistent with improved thermal stability of the BV2365 and NXV-CoV2373 compared to that of WT spike protein (Table 1).

**Transmission electron microscopy and two-dimensional class averaging.** Transmission electron microscopy (TEM) and two-dimensional (2D) class averaging were used to determine the ultrastructure of NVX-CoV2373. High magnification (67,000× and 100,000×) TEM images of negatively stained NVX-CoV2373 showed particles corresponding to S-protein homotrimers. An automated picking protocol supplemented with manual picking was used to construct 2D class average images[17,18]. Two rounds of 2D class averaging of 28,623 homotrimeric structures revealed a triangular particle appearance with a 15-nm length and 13 nm width (Fig. 1c). Overlaying the recently solved cryoEM structure of the SARS-CoV-2 S protein ectodomain (EMD ID: 21374)[19,20] over the 2D NVX-CoV2373 image showed a good fit with the crown-shaped S1 (NTD and RBD) and the S2 stem (Fig. 1c). Also apparent in the 2D images was a faint projection that protruded from the tip of the trimeric structure opposite of the NTD/RBD crown (Fig. 1c). 2D class averaging using a larger box size showed these faint projections form a connection between the S-trimer and an amorphous structure. We speculate these faint projections likely represent the HR2 domain, which is highly flexible in the prefusion conformation[19], with the TM domain anchored within a polysorbate 80 micelle (Fig. 1c).

**SARS-CoV-2 S-protein binding to hACE2 receptor by bio-layer interferometry and ELISA.** S-protein binding to the hACE2 receptor was determined using bio-layer interferometry (BLI). To assess binding, a histidine-tagged hACE2 receptor was coupled to nickel charged nitrilotriacetic acid (Ni-NTA) biosensor tips. The hACE2-coated biosensor tips were dipped in wells containing serially diluted (4.7 nM to 300 nM) recombinant S protein. Dissociation kinetics showed that the S proteins remained tightly bound as evident by minimal or no dissociation over 900 s of observation in the absence of fluid-phase S protein (Fig. 2a–c).

We next determined the specificity of receptor binding using an ELISA method. In this evaluation, histidine-tagged hACE2 or hDDP4 receptors over a concentration range of 0.0001–5 µg mL$^{-1}$ were added to ELISA plates coated with WT, BV2365, or NVX-CoV2373 and binding was detected with HRP-conjugated anti-histidine antibody. WT, BV2365, and NVX-CoV2373 proteins

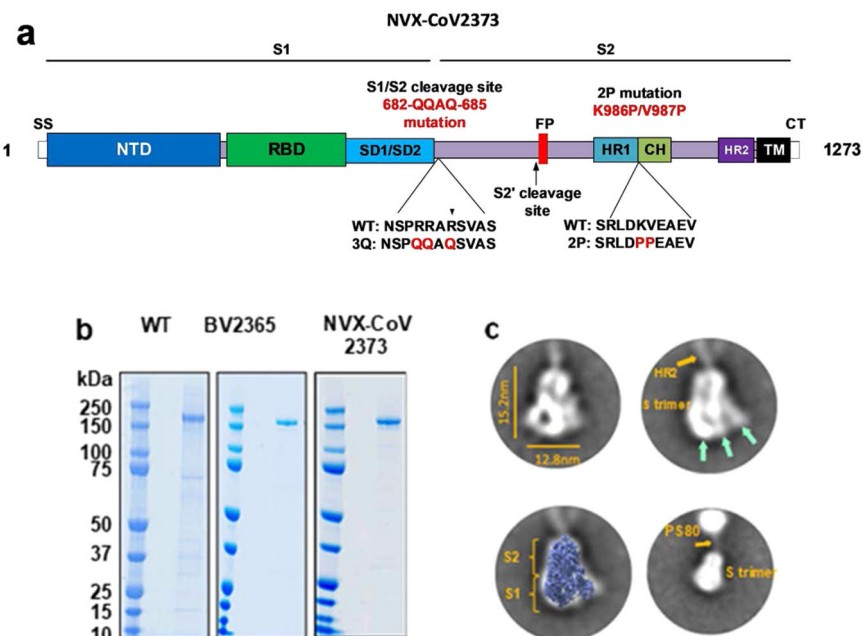

**Fig. 1 SARS-CoV-2 spike glycoprotein constructs. a** Linear diagram of the full-length SARS-CoV-2 spike (S) protein showing the S1 and S2 subunits. Structural elements include a cleavable signal sequence (SS, white), N-terminal domain (NTD, dark blue), receptor binding domain (RBD, green), subdomains 1 and 2 (SD1/SD2, light blue), fusion peptide (FP, red), heptad repeat 1 (HR1, yellow), central helix (CH, light green), heptad repeat 2 (HR2, purple), transmembrane domain (TM, black), and cytoplasmic tail (CT, white). The native furin cleavage site was mutated (RRAR→QQAQ) to be protease resistant to generate the full-length BV2365 variant. BV2365 was further stabilized by introducing two proline (2P) substitutions at positions K986P and V987P to produce the double mutant NVX-CoV2373. **b** Reduced SDS-PAGE gel of purified full-length wild-type (WT), BV2365, and NVX-CoV2373 (representative of 3 to 10 lots). WT spike is produced as a mixture of cleaved and uncleaved proteins. Figure shows purified uncleaved WT spike protein. **c** Transmission electron microscopy and 2D class averaging of NVX-CoV2373. 2D images were constructed from 28,623 NVX-CoV2373 particles followed by two rounds of 2D averaging. 2D images of NVX-CoV2373 S-trimers showing well-defined triangle-shaped particles with a length of 15 nm and a width of 12.8 nm. The S1 apical surface with the N-terminal receptor and receptor-binding domain (NTD/RBD) is indicated by green arrows. Faint protrusions (orange arrow) extending from the tip of the trimers were evident and appear to correspond to the S2 HR2 domain. Class average images showing a good fit of NVX-CoV2373 S-trimer with cryoEM solved structure of the SARS-CoV-2 trimeric spike protein ectodomain (EMD ID: 21374) overlaid on the 2D image. 2D class averaging using a larger box size showing 2D class average image with the less well-defined HR2 (orange arrow) anchoring the S-trimer to polysorbate 80 (PS80) micelle by the C-terminal TM. Source data are provided as a Source Data file.

**Table 1 Particle size and thermostability of SARS-CoV-2 trimeric spike proteins.**

| SARS-CoV-2 S-proteins | Differential scanning calorimetry (DSC) | | Dynamic light scattering | |
|---|---|---|---|---|
| | $T_{max}$ (ºC) | $\Delta H$cal (kJ mol$^{-1}$) | $Z$-avg diameter (nm) | PDI |
| Wild-type | 58.6 | 153 | 69.53 | 0.46 |
| BV2365 | 61.3 | 466 | 33.40 | 0.25 |
| NVX-CoV2373 | 60.4 | 732 | 27.21 | 0.29 |

*$T_{max}$ melting temperature, $Z$-avg Z-average particle size, PDI polydispersity index.*

specifically bound hACE2 but failed to bind the hDPP4 receptor used by MERS-CoV (IC$_{50}$ > 5000 ng mL$^{-1}$). WT and BV2365 bound to hACE2 with similar affinity (IC$_{50}$ = 36–38 ng mL$^{-1}$), while NVX-CoV2373 attained 50% saturation of hACE2 binding at a twofold lower concentration (IC$_{50}$ = 18 ng mL$^{-1}$) (Fig. 2d–f).

**SARS-CoV-2 S stability under stressed conditions**. The stability of a COVID-19 vaccine for global distribution is critical. The structural integrity of the NVX-CoV2373 spike protein with the two proline substitutions and BV2365 without the two proline substitutions was assessed with different environmental stress conditions using the hACE2 ELISA. Incubation of NVX-CoV2373

at pH extremes (48 h at pH 4 and pH 9), with prolonged agitation (48 h), through freeze/thaw (two cycles), or elevated temperature (48 h at 25 °C and 37 °C) had no effect on hACE2 receptor binding (IC$_{50}$ = 14.0–18.3 ng mL$^{-1}$). Only oxidizing conditions with hydrogen peroxide reduced the binding of NVX-CoV2373 by eightfold (IC$_{50}$ = 120 ng mL$^{-1}$) (Fig. 3a). BV2365 without the two proline substitutions was less stable as determined by a significant reduction in hACE2 binding (IC$_{50}$ = 56.8–143.4 ng mL$^{-1}$) under multiple conditions (Fig. 3b). These results confirmed that NVX-CoV2373 with the two proline mutations had significantly greater stability and was therefore selected for further evaluation.

**NVX-CoV2373 vaccine immunogenicity in mice**. We assessed the immunogenicity of NVX-CoV2373 and the dose-sparing potential of saponin-based Matrix-M adjuvant. Groups of mice were immunized with a dose range (0.01 μg, 0.1 μg, 1 μg, and 10 μg) of NVX-CoV2373 with 5 μg Matrix-M adjuvant using a single priming dose or a prime/boost regimen spaced 14 days apart (Fig. 4a). Animals immunized with a single priming dose of 0.1–10 μg NVX-CoV2373/Matrix-M had elevated anti-S IgG titers that were detected 21–28 days after a single immunization (Fig. 4b). Mice immunized with 10 μg NVX-CoV2373/Matrix-M induced antibodies that blocked hACE2 receptor binding to S-protein and virus neutralizing antibodies 21–28 days after a single priming dose (Fig. 4c, d). Animals immunized with the prime/boost regimen had significantly elevated anti-S IgG titers that were detected 7–16 days following the booster immunization

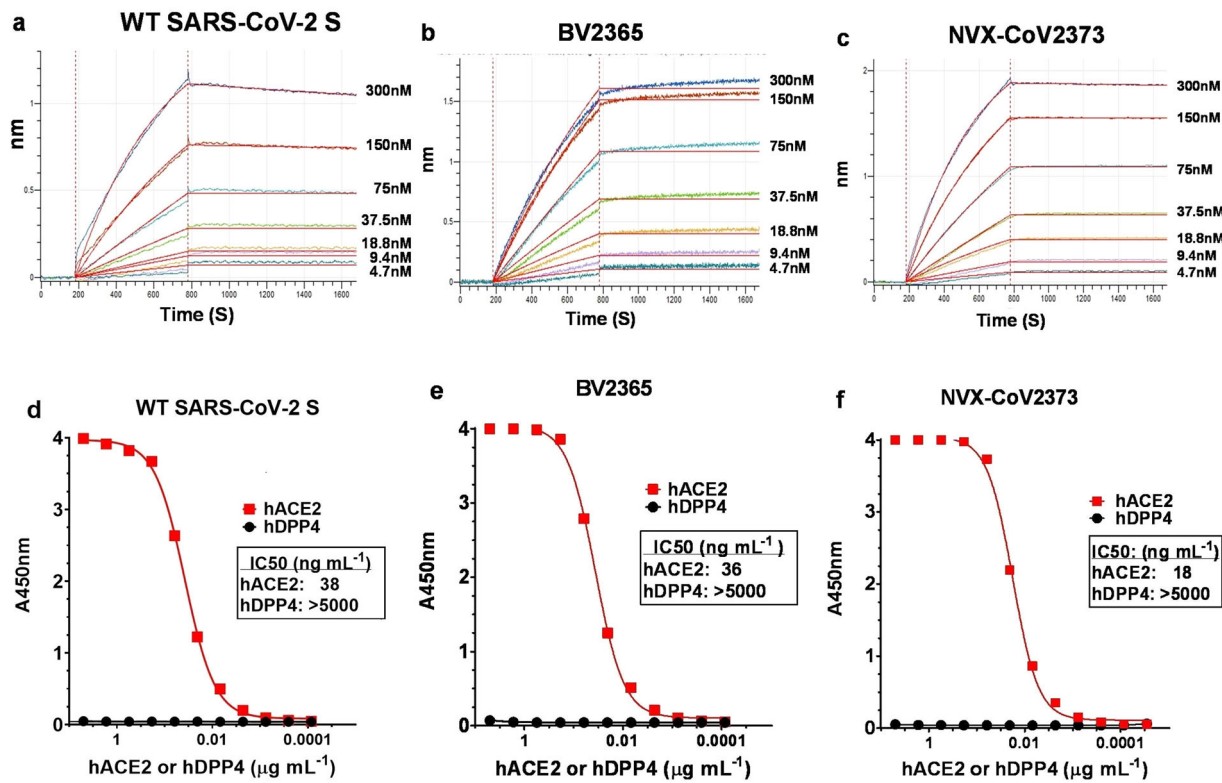

**Fig. 2 Kinetics and specificity of SARS-CoV-2 S protein binding to hACE2 receptor determined by bio-layer interferometry (BLI) and ELISA.** BLI sensorgram showing the binding of **a** wild-type (WT) SARS-CoV-2 S, **b** BV2365, and **c** NVX-CoV2373 spike proteins to histidine-tagged hACE2 receptor immobilized on a Ni-NTA biosensor tip. Data are shown as colored lines at different concentrations of spike protein. Red lines are the best fit of the data. **d** WT-SARS-CoV-2 S, **e** BV2365, and **f** NVX-CoV2373 demonstrated binding to hACE2 receptor but failed to bind hDPP4 as determined by ELISA. Source data are provided as a Source Data file.

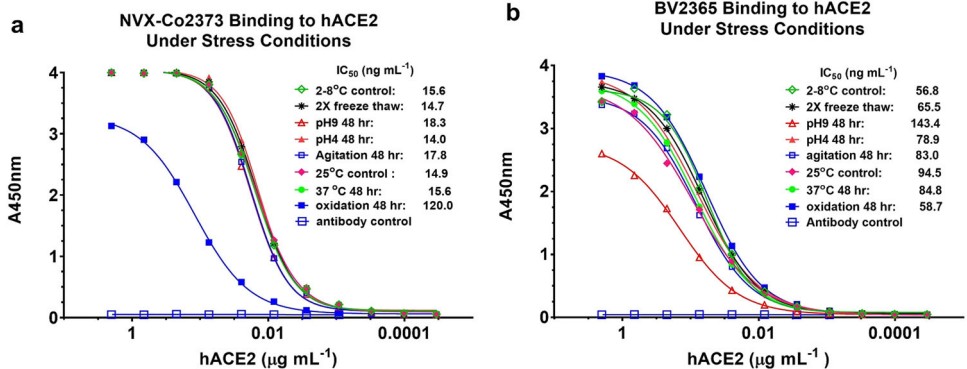

**Fig. 3 Stability of SARS-CoV-2 variants under stress conditions.** The hACE2 receptor binding ELISA method was used to assess the structural integrity of BV2365 and NVXCoV2373 under stressed conditions. **a** NVXCoV2373 and **b** BV2365 were exposed to repeat freeze–thaw cycles, pH extremes, agitation, elevated temperatures, and oxidation for extended periods as indicated. Treated samples were immobilized on 96-well plates, then incubated with serially diluted (2–0.0001 µg mL$^{-1}$) histidine-tagged hACE2. Bound receptor was detected with HRP-conjugated rabbit anti-histidine IgG. Source data are provided as a Source Data file.

across all dose levels. Animals immunized with 1 µg and 10 µg NVX-CoV2373/Matrix-M had similar high anti-S IgG titers following immunization (geometric mean titer, GMT = 139,000 and 84,000, respectively). Importantly, mice immunized with 0.1 µg, 1 µg, or 10 µg NVX-CoV/Matrix-M had significantly ($p \leq 0.00006$) higher anti-S IgG titers compared to mice immunized with 10 µg NVX-CoV2373 without adjuvant (Fig. 4b). These results indicate the potential for a tenfold or greater dose sparing provided by Matrix-M adjuvant. Furthermore, immunization with two doses of NVX-CoV2373/Matrix-M elicited high titer antibodies that blocked hACE2 receptor binding to S-protein (IC$_{50}$ = 218–1642)

and neutralized the cytopathic effect (CPE) of SARS-CoV-2 on Vero E6 cells (100% blocking of CPE = 7680–20,000) across all dose levels (Fig. 4c, d).

**NVX-CoV2373 protection against SARS-CoV-2 in Ad/hACE2 mice.** Mice were vaccinated with NVX-CoV2373 to evaluate the induction of protective immunity against challenge with SARS-CoV-2 by comparing single-dose or prime/boost vaccination strategies in a live virus challenge model. Mice were immunized with a single priming dose or a prime/boost regimen with NVX-CoV2373/Matrix-M as described above. Since mice do not

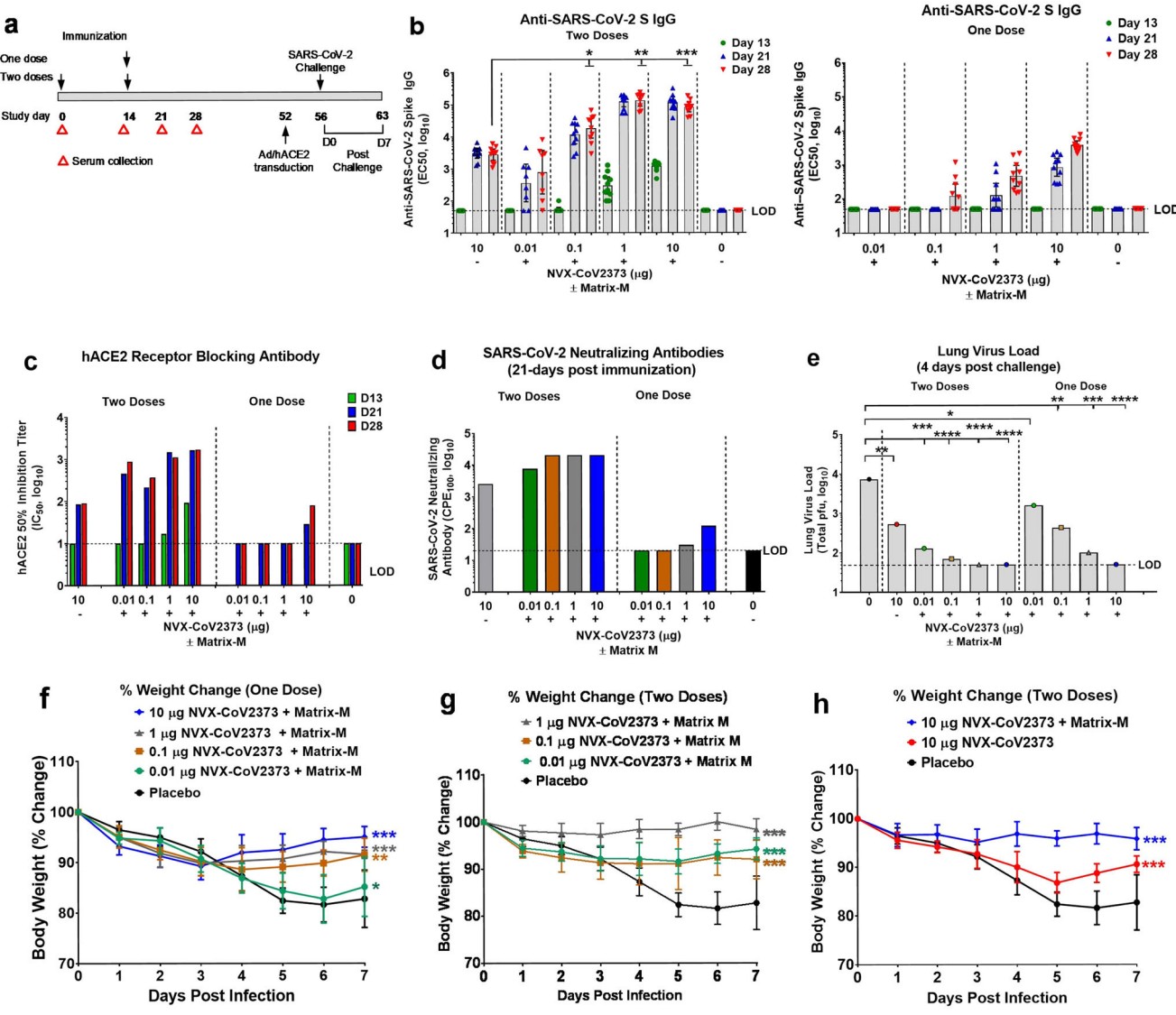

**Fig. 4 Immunogenicity of NVX-CoV2373 vaccine and protection against SARS-CoV-2 infection in mice. a** Groups of mice ($n = 10$/group) were immunized with a single priming dose (study day 14) or a prime/boost spaced 14 days apart (study days 0 and 14) with a dose range of NVX-CoV2373 with Matrix-M adjuvant (5 μg). A control group received formulation buffer (placebo). **b** Anti-SARS-CoV-2 IgG titers. Bars indicate the geometric mean titer (GMT) and the error bars indicate the 95% CI for each immunization group ($n = 10$/group). Individual animal values are indicated by colored symbols. Comparison of anti-SARS-CoV2 S IgG titer between a group immunized with two doses of NVX-CoV2373 (10 μg) without adjuvant compared to groups receiving different dose levels (0.01, 0.1, 1, and 10 μg) of NVX-CoV2373 with adjuvant. Comparisons were performed by Student's t-test (unpaired, two tail); \*$p = 5.6\text{E}{-}05$, \*\*$p = 2.4\text{E}{-}11$, \*\*\*$p = 2.8\text{E}{-}11$. **c** hACE2-receptor-blocking antibodies in pooled serum ($n = 10$/group) collected after the first immunization (study days 13, 21, and 28). Bars indicate the mean of replicate assays. **d** SARS-CoV-2 virus-neutralizing antibody titers in pooled serum ($n = 10$/group) collected from groups receiving a single dose or a prime/boost. Bars indicate the mean of replicate assays. Following the booster immunization (study day 52), mice were transduced intranasally with $2.5 \times 10^8$ pfu Ad/CMVhACE2. At 4 days post transduction (study day 56), mice were challenged intranasal with $1.5 \times 10^5$ pfu of SARS-CoV-2. Animals were monitored daily for up to 7 days post infection (D0–D7). **e** Infectious virus load in lung homogenates at 4 days post SARS-CoV-2 challenge (D4). Bars represent the mean virus load ($n = 5$/group). Individual animal values are indicated by colored symbols. Comparisons were performed by Student's t-test (unpaired, two tail); \*$p = 0.02$, \*\*$p \leq 0.003$, \*\*\*$p \leq 1.0\text{E}{-}05$, \*\*\*\*$p \leq 4.0\text{E}{-}06$. Weight change was determined for up to 7 days following nasal challenge with SARS-CoV-2 (study days 56–63). **f** Mice immunized with one dose. **g, h** Mice immunized with two doses. Results are plotted as the mean and the error bars indicate the ±SD (D0–D4 $n = 10$ mice/time point and D5–D7 $n = 5$ mice/time point). Two-way ANOVA was used to compare differences in weight change of vaccinated groups compared to the placebo control group; \*$p = 0.5$ (not significant), \*\*$p = 0.001$, \*\*\*$p \leq 0.0001$. Dashed black line indicates the limit of detection (LOD). Source data are provided as a Source Data file.

support replication of WT SARS-CoV-2 virus, BALB/c mice were transduced with adenovirus encoding human ACE2 receptor (Ad/hACE2) which renders them permissive to infection with SARS-CoV-2 (refs. [21,22]). At 4 days post transduction, mice were challenged with $10^5$ plaque forming units (pfu)/mouse of SARS-

CoV-2 (WA1 strain). Following challenge, mice were weighed daily and pulmonary histology and viral load were analyzed at 4 and 7 days post challenge.

At 4 days post infection (dpi), placebo-treated mice had an average of $10^4$ SARS-CoV-2 pfu/lung, while the mice immunized

with NVX-CoV2373 without Matrix-M had $10^3$ pfu/lung and those with Matrix-M had limited to no detectable virus load (Fig. 4e). The NVX-CoV2373 with Matrix-M prime-only groups of mice exhibited a dose-dependent reduction in virus titer, with recipients of the 10 μg dose having no detectable virus at day 4 post infection. Mice receiving 1 μg, 0.1 μg, and 0.01 μg doses all showed a marked reduction in titer compared to placebo-vaccinated mice. In the prime/boost groups, mice immunized with 10 μg, 1 μg, and 0.1 μg doses had almost undetectable lung virus loads, while the 0.01 μg group displayed a reduction of at least 1 log relative to placebo animals. Weight loss during the experiment paralleled the viral load findings, with animals receiving single dose of 10 μg, 1 μg, and 0.1 μg NVX-CoV2373/Matrix-M showing marked protection from weight loss compared to the unvaccinated placebo animals (Fig. 4f). Mice receiving a prime and boost vaccination with adjuvanted vaccine also demonstrated significant ($p \le 0.0001$) protection against weight loss at all dose levels (Fig. 4g). In addition, we compared the prime/boost regimens using 10 μg of either adjuvanted or unadjuvanted NVX-CoV2373. The mice receiving the prime/boost with adjuvant were significantly protected from weight loss relative to placebo mice, while the group immunized with 10 μg NVX-CoV2373 alone were partially protected against weight loss (Fig. 4h). These results confirmed that NVX-CoV2373 confers protection against SARS-CoV-2 and that low doses of the vaccine associated with lower serologic responses do not exacerbate weight loss or induce exaggerated illness.

**Histopathology**. Lung histopathology was evaluated on days 4 and 7 post challenge. At day 4 post challenge, placebo-immunized mice showed denudation of epithelial cells in the large airways with thickening of the alveolar septa surrounded by a mixed inflammatory cell population. Perivascular cuffing was observed throughout the lungs with inflammatory cells consisting primarily of neutrophils and macrophages. By day 7 post infection, the placebo-treated mice displayed peribronchiolar inflammation with increased perivascular cuffing. The thickened alveolar septa remained, with increased diffuse interstitial inflammation throughout the alveolar septa (Fig. 5). Histopathology scores also show the level of peribronchiolar and perivascular inflammation in lungs of immunized mice was significantly reduced ($p \le 0.002$) compared to the placebo group (Supplementary Fig. 1). Importantly, these data demonstrated that NVX-CoV2373 reduces lung inflammation after challenge and that doses and regimens of NVX-CoV2373 that elicit minimal or no detectable neutralizing activity are not associated with any obvious exacerbation of the inflammatory response to the virus.

The NVX-CoV2373-immunized mice showed a significant reduction in lung pathology at both 4 and 7 dpi in a dose-dependent manner. The prime only group displayed reduced inflammation at the 10 μg and 1 μg doses with a reduction in inflammation surrounding the bronchi and arterioles compared to placebo mice. In the lower doses of the prime-only groups, lung inflammation resembled that of the placebo groups, correlating with weight loss and lung virus titer. The prime/boost immunized groups displayed a significant reduction in lung inflammation for all doses tested (Supplementary Fig. 1), which again correlated with lung viral titer and weight loss data (Fig. 4e–h). The epithelial cells in the large and small bronchi at days 4 and 7 were substantially preserved with minimal bronchiolar sloughing or signs of viral infection compared to

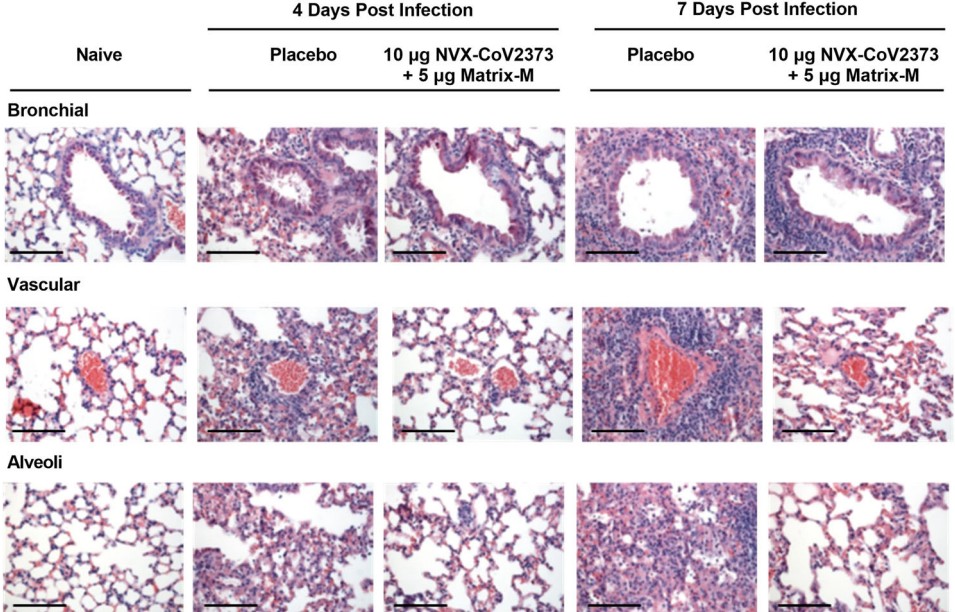

**Fig. 5 Histopathological analysis of SARS-CoV-2 infection in NVX-CoV2373-immunized mice transduced with Ad/hACE2 and challenged with SARS-CoV-2.** Groups of mice (N = 10/group) were immunized with NVX-CoV2373 with or without Matrix-M (5 μg) with two doses spaced 14 days apart. Placebo group received formulation buffer. Following immunization, mice were intranasally transduced with Ad/CMV/hACE2, 52 days after the first priming dose. At 4 days post transduction, mice were challenged with 1 × $10^5$ pfu/mouse of SARS-CoV-2 (WA1 strain). Lungs were collected 4 and 7 days post infection. Representative placebo control animal at 4 days post infection showing denuding of bronchial epithelium with marked thickening of the alveolar septa surrounded by mixed inflammatory cells. Diffuse perivascular cuffing was observed throughout the lung, consisting of neutrophils and macrophages. At 7 days post infection, peribronchiolar inflammation and perivascular cuffing was markedly increased. Lungs from NVX-CoV2373-vaccinated animals had little or no epithelial cell sloughing or infection within large and small bronchi at days 4 and 7 post infection. There was no evidence of exacerbated lung inflammation in NVX-CoV2373-immunized animals. Images from age- and sex-matched BALB/c mice from a separate experiment that were transduced with Ad/hACE2 alone as a control are shown in the right panel at 7 days post transduction. Scale bar 100 μm.

placebo at 7 dpi (Mean score = 1.8 vs 3.8). The arterioles of animals immunized with 10 µg, 1 µg, and 0.1 µg doses had minimal inflammation with only moderate cuffing seen in the 0.01 µg dose, similar to placebo at 7 dpi (Mean score = 1.8 vs 4.6, respectively) (Supplementary Fig. 1). Alveolar inflammation was reduced in animals that received the higher doses with only the lower 0.01 µg dose with inflammation (Fig. 5). These data demonstrated that NVX-CoV2373 reduces lung inflammation after SARS-CoV-2 challenge and that doses and regimens of NVX-CoV2373 that elicit minimal or no detectable neutralizing activity are not associated with any obvious exacerbation of the inflammatory response to the virus. Reduced pulmonary inflammation, reduced or absence of virus load in the lungs (Fig. 4e), and protection against weight loss (Fig. 4f, g, h) were consistent with vaccine-induced protection against virus challenge in Ad/hACE2-transduced mice.

**Multifunctional cytokine analysis of CD4$^+$ and CD8$^+$ T cells in mice.** To determine the role of Matrix-M in generating T cell responses, we immunized groups of mice ($N = 6$/group) with 10 µg NVX-CoV2373 alone or with 5 µg Matrix-M in a two-dose regimen spaced 21 days apart. Antigen-specific T cell responses were measured by ELISpot and intracellular cytokine staining (ICCS) from spleens collected 7 days after the second immunization (study day 28) (Fig. 6a). The number of IFN-γ-secreting cells after ex vivo stimulation increased sevenfold in spleens of mice immunized with NVX-CoV2373/Matrix-M compared to NVX-CoV2373 alone as measured by the ELISpot assay (Fig. 6b). In order to examine CD4$^+$ and CD8$^+$ T cell responses separately, ICCS assays were performed in combination with surface marker staining. Data shown were gated on the CD44$^{hi}$ CD62L$^-$ effector memory T cell population. Importantly, we found the frequency of IFN-γ$^+$, TNF-α$^+$, and IL-2$^+$ cytokine-secreting CD4$^+$ and CD8$^+$ T cells was significantly higher ($p < 0.0001$) in spleens from the NVX-CoV2373/Matrix-M-immunized mice compared to mice immunized without adjuvant (Fig. 6c, d). Further, we noted the frequency of multifunctional CD4$^+$ and CD8$^+$ T cells, which simultaneously produce at least two or three cytokines, was also significantly increased ($p < 0.0001$) in spleens from the NVX-CoV2373/Matrix-M-immunized mice (Fig. 6c, d). Immunization with NVX-CoV2373/Matrix-M resulted in higher proportions of a multifunctional phenotype within both CD4$^+$ and CD8$^+$ T cell populations. The proportions of multifunctional phenotypes detected in memory CD8$^+$ T cells were higher than those in CD4$^+$ T cells (Fig. 6e).

Type 2 cytokines IL-4 and IL-5 secretion from CD4$^+$ T cells was also determined by ICCS and ELISpot, respectively. We found that immunization with NVX-CoV2373/Matrix-M also increased type 2 cytokines IL-4 and IL-5 secretion (twofold) compared to immunization with NVX-CoV2373 alone, but to a lesser degree than enhancement of type 1 cytokine production (e.g. IFN-γ increased 20-fold). These results indicate that administration of the Matrix-M adjuvant led to an antigen-specific CD4$^+$ T cell development, which was at least balanced between Th1 and Th2 phenotypes or, in most animals, was Th1-dominant (Supplementary Fig. 2).

Having shown that vaccination with NVX-CoV2373/Matrix-M elicited multifunctional, antigen-specific, CD4$^+$ T cell responses and virus-neutralizing antibodies in mice, we next evaluated the effect of the immunization on germinal center (GC) formation by measuring the frequency of CD4$^+$ T follicular helper (Tfh) and GC B cells in spleens. Addition of Matrix-M adjuvant significantly increased the frequency of Tfh cells (CD4$^+$ CXCR5$^+$ PD-1$^+$) ($p = 0.01$), as well as the frequency of GC B cells (CD19$^+$GL7$^+$CD95$^+$) ($p = 0.0002$) in spleens (Fig. 7a, b).

**Immunogenicity of NVX-CoV2373 vaccine in olive baboons.** Having determined that low doses of NVX-CoV2373 with Matrix-M elicit protective neutralizing antibodies and promote the generation of multifunctional antigen-specific T cells in mice, we next evaluated the immunogenicity of the vaccine in adult baboons. In this study, adult olive baboons were immunized with a dose range (1 µg, 5 µg, and 25 µg) of NVX-CoV2373 with 50 µg Matrix-M adjuvant administered by IM injection in two doses spaced 21 days apart (Fig. 8a). To assess the adjuvant activity of Matrix-M in nonhuman primates, an additional group of animals was immunized with 25 µg of NVX-CoV2373 without the adjuvant. Anti-S protein IgG titers were detected within 21 days of a single priming immunization in animals immunized with NVX-CoV2373/Matrix-M across all the dose levels (GMT = 1249–19,000). Anti-S protein IgG titers increased over 1 log (GMT = 33,000–174,000) within 1–2 weeks following a booster immunization (days 28 and 35) across all dose levels. Importantly, animals immunized with NVX-CoV2373 without adjuvant had minimal or no detected anti-S IgG titer (GMT <125) after one immunization, which was not boosted by a second immunization (Fig. 8b).

We also determined the functionality of the antibodies. Low levels of hACE2-receptor-blocking antibodies were detected in animals following a single immunization with 5 or 25 µg NVX-CoV2373/Matrix-M (GMT = 22–37). Receptor-blocking antibody titers were significantly increased within 1–2 weeks of the booster immunization across all groups immunized with NVX-CoV2373/Matrix-M (GMT = 150–600) (Fig. 8c). Virus-neutralizing antibodies were also elevated (GMT = 190–446) across all dose groups after a single immunization with NVX-CoV2373/Matrix-M. Animals immunized with 25 µg of NVX-CoV2373 alone had no detectable antibodies that block S-protein binding to hACE2 (Fig. 8c). Neutralizing titers increased 25- to 38-fold following the second immunization (GMT = 6400–17,000) (Fig. 8d). Animals receiving NVX-CoV2373 alone had little or no detectable neutralizing antibodies (GMT <100). There was a significant correlation ($p < 0.0001$) between anti-S IgG levels and neutralizing antibody titers (Fig. 8e). The immunogenicity of the adjuvanted vaccine in nonhuman primates is consistent with the mouse immunogenicity results and further supports the role of Matrix-M adjuvant in promoting the generation of neutralizing antibodies and dose sparing.

PBMCs were collected 7 days after the second immunization (day 28) and T cell response was measured by ELISpot assay. PBMCs from animals immunized with 5 µg or 25 µg NVX-CoV2373/Matrix-M had the highest number of IFN-γ secreting cells, which was fivefold greater on average compared to animals immunized with 25 µg NVX-CoV2373 alone or 1 µg NVX-CoV2373/Matrix-M (Fig. 8f). By ICCS analysis, immunization with 5 µg NVX-CoV2373/Matrix-M also showed the highest frequency of IFN-γ$^+$, IL-2$^+$, and TNF-α$^+$ CD4$^+$ T cells (Fig. 8g). This trend was also true for multifunctional CD4$^+$ T cells, in which at least two or three type 1 cytokines were produced simultaneously (Fig. 8h). Type 2 cytokine IL-4 levels were too low to be detected in baboons by ELISpot analysis.

We next compared the levels of serum antibodies in recovered COVID-19 patients to the level of antibodies in NVX-CoV2373/Matrix-M vaccinated baboons. The mean anti-S IgG levels were sevenfold higher in immunized baboons (EC$_{50}$ = 152,060, 95%CI, 60,767–243,354) compared to convalescent serum (EC$_{50}$ = 21,136, 95%CI, 11,473–30,799). We also compared the level of functional hACE2-receptor-inhibiting (50% RI) titers. Baboons receiving the vaccine had eightfold higher binding and receptor inhibiting antibodies (50% RI = 478, 95%CI, 161.1–794.4) compared to levels in COVID-19 convalescent serum (50% RI = 61, 95%CI, 35.7–85.5) (Fig. 9). Therefore, NVX-CoV2373 vaccine induced binding and functional antibodies in a nonhuman primate at levels comparable or higher than individuals recovered from COVID-19. Collectively,

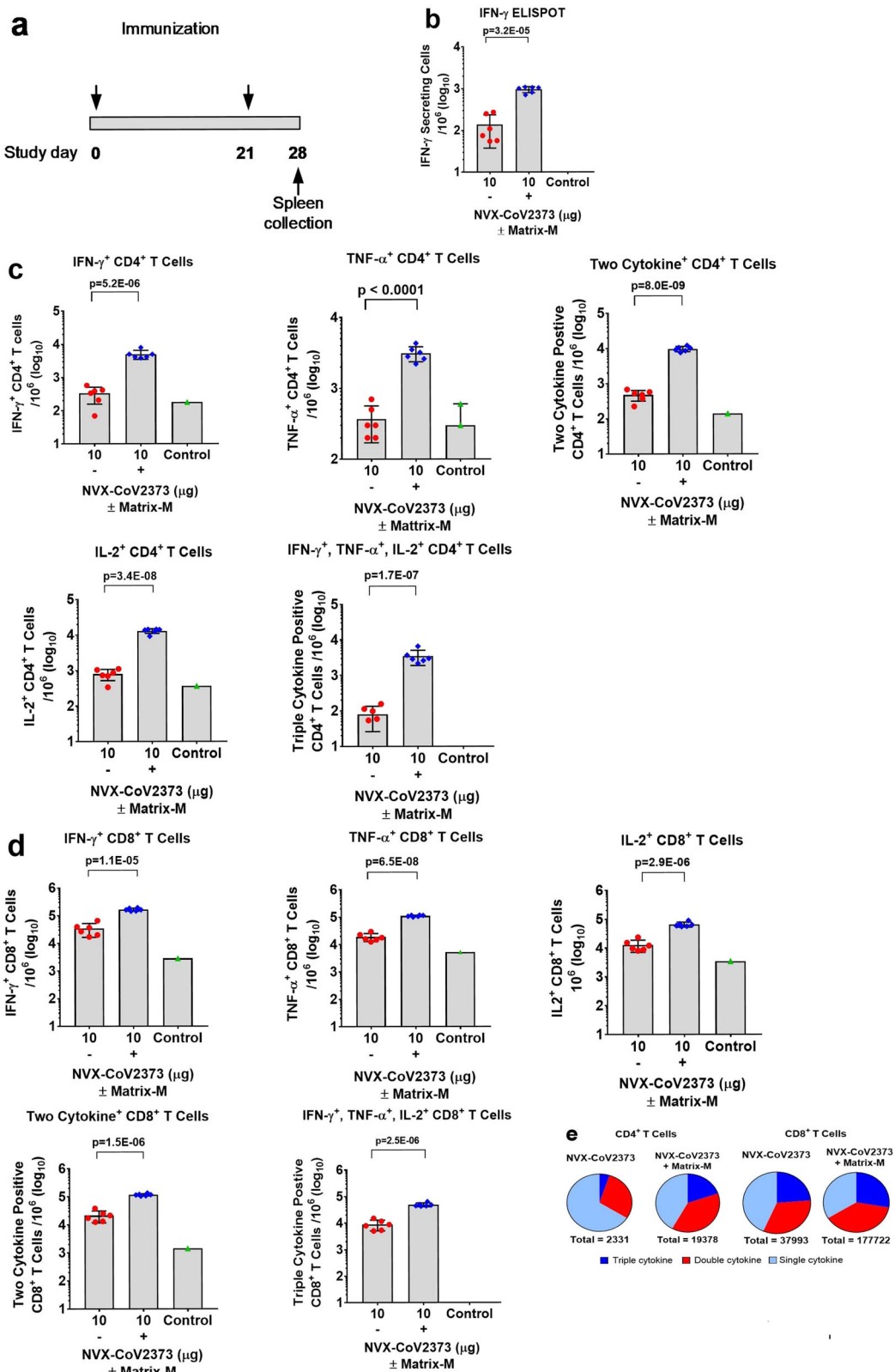

these results support the development of NVX-CoV2373 for prevention of COVID-19.

## Discussion
Here, we showed that a full-length, stabilized prefusion SARS-CoV-2 S glycoprotein vaccine (NVX-CoV2373) adjuvanted by Matrix-M can induce high levels of functional immunity in mice and baboons, and protects mice expressing hACE2 receptors in a live SARS-CoV-2 challenge. The functional immunity induced by this nanoparticle vaccine and Matrix-M adjuvant clearly depends on both the adjuvant and antigen components and mirrors the human experience with influenza hemagglutinin vaccine[23] and a naïve population with Ebola recombinant

**Fig. 6 Multifunctional cytokine analysis of SARS-CoV-2 S-specific CD4+ and CD8+ T cells in immunized mice. a** Groups of mice ($n = 6$/group) were immunized with 10 μg NVX-CoV2373 with and without 5 μg Matrix-M adjuvant in two doses spaced 21 days apart. A negative control group ($N = 3$) was not immunized. Splenocytes were collected 7 days after the second immunization (study day 28) and stimulated with a peptide pool (PP) that covers the entire spike protein for 6 h. **b** The number of IFN-γ secreting cells per million splenocytes was determined by ELISpot ($n = 6$/group). **c, d** The frequency of CD4+ memory T cells and CD8+ memory T cells producing IFN-γ, TNF-α, and IL-2, or at least 2 of 3 cytokines was determined by intracellular cytokine staining ($n = 6$/group). Analyzed cells were gated on the CD44hiCD62L− effector memory population. Bars represent the mean and the error bars indicate ±SD of triplicate assays. Individual animal values are indicated by colored symbols. Comparisons between groups receiving NVX-CoV2373 with and without adjuvant was performed by Student's $t$-test (unpaired, two tail). **e** Pie charts represent the relative proportion of CD4+ and CD8+ T cells producing one, two, or three cytokines (IFN-γ, TNF-α, and IL-2) in mice immunized with NVX-CoV2373 antigen with and without Matrix-M. Source data are provided as a Source Data file.

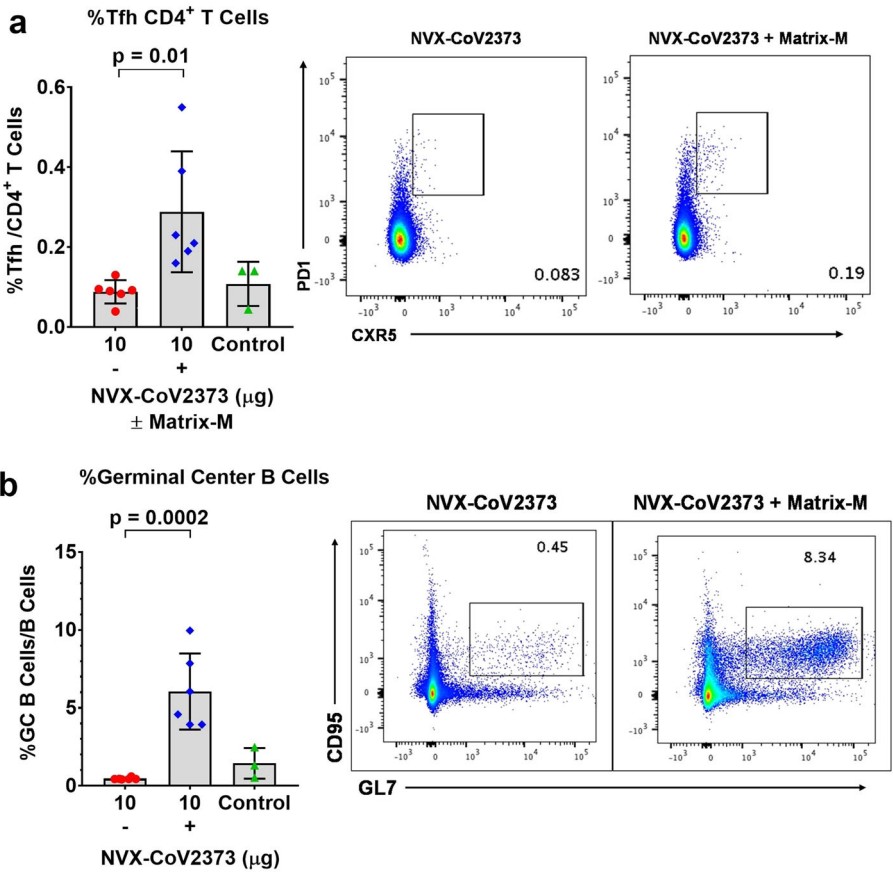

**Fig. 7 Frequencies of follicular helper T cell (Tfh) and germinal center (GC) B cells generated by immunization with NVX-CoV2373 and Matrix-M adjuvant.** Mice were immunized with NVX-CoV2373 with and without Matrix-M adjuvant and splenocytes were collected 7 days after the second immunization. **a** The frequency of splenic Tfh cells (CXCR5+ PD-1+ CD4+) in the CD4 T population. **b** The frequency of splenic germinal center (GC) B cells (GL7+ CD95+ CD19+) in B cells. Bars represent the mean values and the error bars indicate ±SD of triplicate assays. Colored symbols indicate individual animal values. Comparisons between groups receiving NVX-CoV2373 with and without adjuvant was performed by Student's $t$-test (unpaired, two tail). Source data are provided as a Source Data file.

protein nanoparticle vaccines[24]. The mechanism of action of the saponin-based Matrix-M adjuvant has been described by Reimer et al.[25], and further characterized by Magnusson et al.[26]. Subcutaneous injection of Matrix-M has been shown to promote the recruitment of leukocytes to local draining lymph nodes and the spleen in mice, along with an increase in activation marker CD69 on monocytes and T-, B-, natural killer, and dendritic cells in the draining lymph nodes. It is hypothesized that Matrix-M induces a local transient pro-inflammatory response with recruitment, activation, and maturation of immune cells, specifically stimulating dendritic cell uptake and processing of antigens with enhanced presentation[25–27]. In this study,

immunization with NVX-CoV2373 and Matrix-M in mice and nonhuman primates induced anti-S antibodies, hACE2-receptor-inhibiting antibodies, and SARS-CoV-2-neutralizing antibodies after one dose with significantly increased titers after a booster immunization. In addition, NVX-CoV2373 vaccine induced CD4+ and CD8+ T cell responses, and in mice provided protection against SARS-CoV-2 challenge. Matrix-M adjuvant was also shown to enhance Tfh cell and GC B cell development, which are critical for inducing and maintaining a high affinity antibody response. There was no evidence of VAERD in challenged mice immunized with low doses of NVX-CoV2373 (refs. [28,29]).

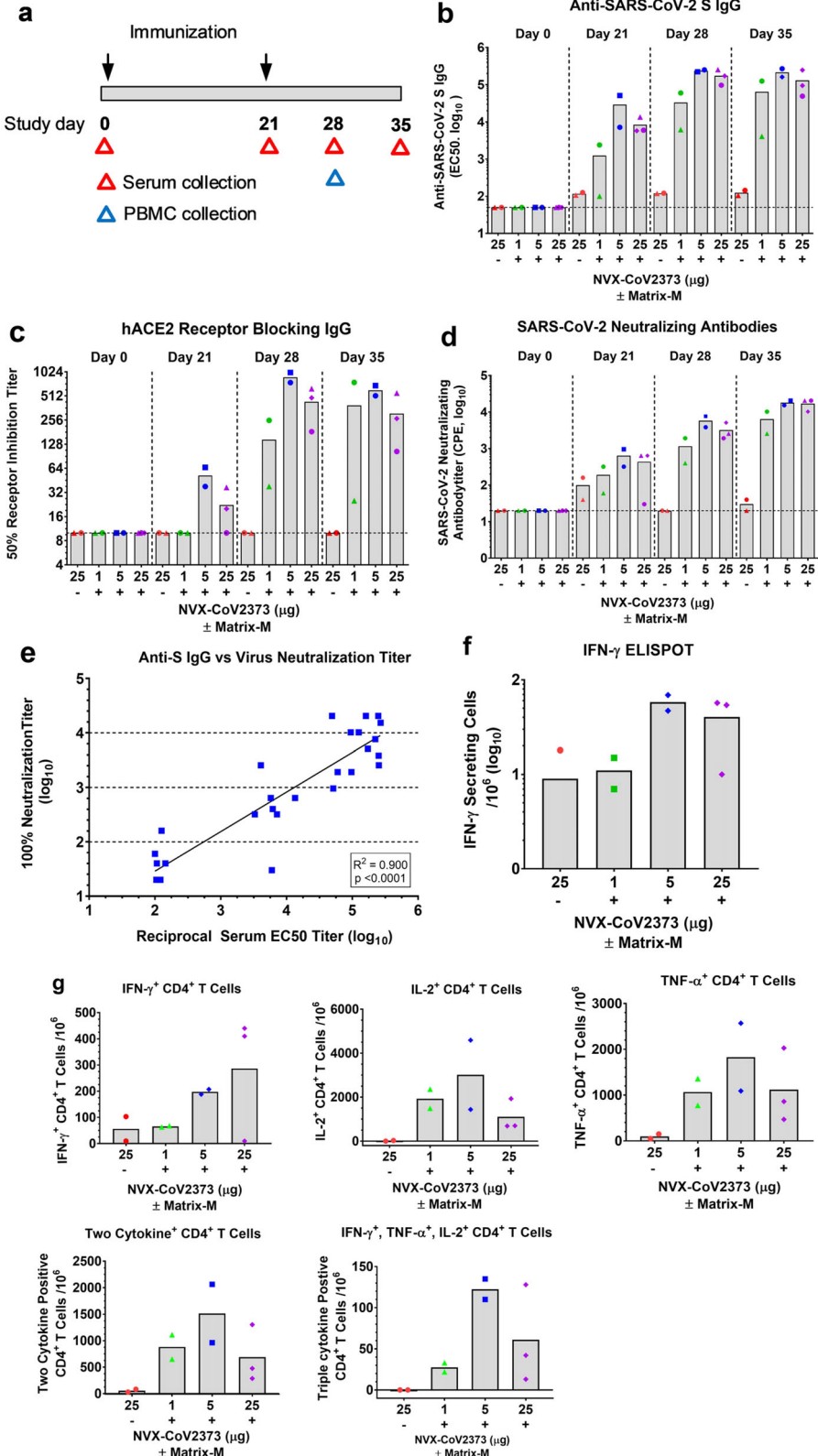

While multiple animal models have been developed for infection with human coronaviruses, including SARS-CoV, MERS-CoV, and now SARS-CoV-2, to date none of them fully represent the pathology or clinical symptoms of human infection. A murine hACE2 transduced challenge model with WT virus is considered robust and was recently used to evaluate protection against SARS-CoV-2 (refs. [21,22]). The distribution and kinetics of ACE2 receptor expression in the bronchiolar epithelium and pneumocytes of transduced mice has been characterized in this model. Of note, the adenovirus-based hACE-2 transduction itself gives rise to some background inflammatory changes which are present in all animals and are, of course, not responsive to prophylaxis

**Fig. 8 Humoral and cellular immune response to NVX-CoV2373 with and without Matrix-M adjuvant in baboons. a** Baboons were randomly assigned to groups ($n = 2$–3/group) and immunized by IM injection with 1, 5, or 25 μg of NVX-CoV2373 and 50 μg Matrix-M adjuvant in two doses spaced 21 days apart (D0 and D21). A separate group ($n = 2$) received two doses of 25 μg NVX-CoV2373 without adjuvant. For serologic analysis, serum was collected prior to immunization (D0) and 21, 28, and 35 days after the first immunization (red triangle). For cellular responses, peripheral blood mononuclear cells (PBMCs) were collected 7 days after the booster (blue triangle) and re-stimulated with purified NVX-CoV2373 spike protein. **b** Anti-SARS-CoV-2 S IgG titers were determined by ELISA. **c** hACE2-receptor-blocking antibodies were determined by ELISA. **d** SARS-CoV-2-neutralizing antibodies determined by in vitro inhibition of cytopathic effect (CPE). Sold bars indicate the group mean and the colored symbols are individual animal values. The horizontal dashed black line indicates the limit of detection (LOD) for each assay. **e** Correlation of anti-SARS-CoV-2 S IgG titers vs SARS-CoV-2-neutralizing antibodies. **f** IFN-γ-secreting PBMCs re-stimulated with NVX-CoV2373 protein were determined by ELISpot anaylsis. **g** Frequency of SARS-CoV-2 spike-specific CD4+ T cells producing single and multiple combinations of type 1 cytokines IFN-γ, TNF-α, and IL-2 determined by intracellular cytokine staining (ICCS). Solid bars represent the group mean and individual animal values are indicated by colored symbols. Source data are provided as a Source Data file.

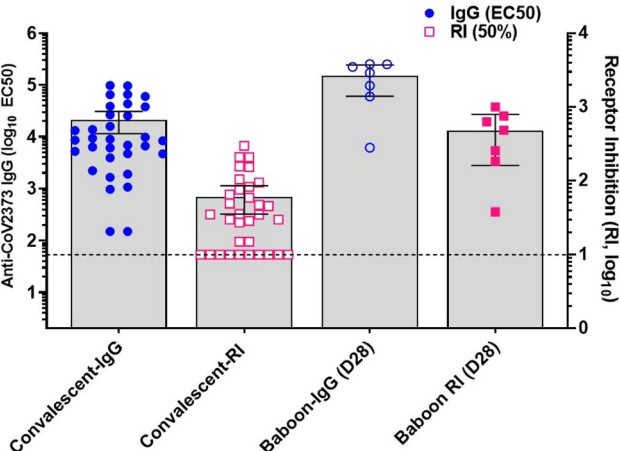

**Fig. 9 Comparison of COVID-19 human convalescent serum antibody levels to NVX-CoV2373-vaccinated baboon antibody levels.**
Convalescent sera were collected from recovered COVID-19 patients 4–6 weeks after testing positive for SARS-CoV-2 ($n = 33$). Sera were analyzed for anti-SARS-CoV-2 S IgG and human ACE2 receptor inhibition antibody levels (50% RI) and antibody levels compared to levels in serum of NVX-CoV2373 with Matrix-M immunized baboons as described in Fig. 8 ($n = 7$). The bars represent the group mean and error bars indicate the 95% confidence interval. The horizontal black dashed line indicates the limit of detection. Source data are provided as a Source Data file.

targeting of SARS-CoV-2. Nonetheless, vaccine-induced functional immunity demonstrated the induction of protective immunity in this study. The safety, immunogenicity, and efficacy of NVX-CoV2373 with Matrix-M adjuvant are currently being evaluated in multiple nonhuman primate models and a phase 1/2 human clinical trial (NCT04368988)[30].

## Methods
**Cell lines, virus, antibody reagents, and receptors**. Vero E6 cells (ATCC, CRL-1586) were maintained in Minimal Eagles Medium (MEM) supplemented with 10% fetal bovine serum, 1% glutamine, and 1% penicillin and streptomycin. The SARS-CoV-2 (WA-1) isolated was obtained from the Center for Disease Control (WA-1 strain) and stock virus prepared in Vero E6 cells. Histidine-tagged hACE2 and histidine-DPP4 receptors were purchased from Sino Biologics (Beijing, CN). Rabbit anti-SARS-CoV S protein was purchased from Biodefense and Emerging Infections Research Resources Repository (catalog no. NR-4569, BEI Resources, Manassas, VA).

**SARS-CoV-2 protein expression**. SARS-CoV-2 constructs were synthetically produced from the full-length S glycoprotein gene sequence (GenBank MN908947 nucleotides 21563–25384). The full-length S-genes were codon optimized for expression in *Spodoptera frugiperda* (Sf9) cells and synthetically produced by GenScript (Piscataway, NJ, USA). The QuikChange Lightning site-directed mutagenesis kit (Agilent) was used to produce two spike protein variants: modifications

were made to the S1/S2 cleavage site by mutating the furin cleavage site (682-RRAR-685) to 682-QQAQ-685 to be protease resistant and designated as BV2365. The single mutant BV2365 was further stabilized by introducing two proline substitutions at positions K986P and V987P (2P) to produce the double mutant, NVX-CoV2373. Full-length S-genes were cloned between the BamHI–HindIII sites in the pBac-1 baculovirus transfer vector (Invitrogen, Carlsbad, CA) under transcriptional control of the *Autographa californica* polyhedron promoter. Recombinant baculovirus constructs were plaque purified and master seed stocks prepared and used to produce the working virus stocks. The baculovirus master and working stock titers were determined using a rapid titer kit (Clontech, Mountain View, CA). Recombinant baculovirus stocks were prepared by infecting Sf9 cells with a multiplicity of infection (MOI) of ≤0.01 pfu per cell[31–33].

**Expression and purification**. SARS-CoV-2 S proteins were produced in Sf9 cells expanded in serum-free medium to a density of $2$–$3 \times 10^6$ cells mL$^{-1}$ and infected with recombinant baculovirus at MOI of ≤0.1 pfu per cell. Cells were cultured at $27 \pm 2$ °C and harvested at 68–72 h post-infection by centrifugation ($3000 \times g$ for 15 min). Cell pellets were suspended in 25 mM Tris HCl (pH 8.0), 50 mM NaCl, and 0.5–1.0% (v/v) TERGITOL NP-9 with leupeptin. S-proteins were extracted from the plasma membranes with Tris buffer containing NP-9 detergent and clarified by centrifugation at $10,000 \times g$ for 30 min. S-proteins were purified by TMAE anion exchange and lentil lectin affinity chromatography. Hollow fiber tangential flow filtration was used to formulate the purified spike protein at 100–150 μg mL$^{-1}$ in 25 mM sodium phosphate (pH 7.2), 300 mM NaCl, 0.02% (v/v) polysorbate 80 (PS 80)[32]. Purified S-proteins were evaluated by 4–12% gradient SDS-PAGE stained with Gel-Code Blue reagent (Pierce, Rockford, IL) and purity was determined by scanning densitometry using the OneDscan system (BD Biosciences, Rockville, MD).

**Dynamic light scattering**. Samples were equilibrated at 25 °C and scattering intensity was monitored as a function of time in a Zetasizer NanoZS v7.12 (Malvern, UK). Cumulants analysis of the scattered intensity autocorrelation function was performed with instrument software to provide the $Z$-average particle diameter and PDI.

**Differential scanning calorimetry**. Samples and corresponding buffers were heated from 4 °C to 120 °C at 1 °C per minute and the differential heat capacity change was measured in a NanoDSC v4.6.0 (TA Instruments, New Castle, DE). A separate buffer scan was performed to obtain a baseline, which was subtracted from the sample scan to produce a baseline-corrected profile. The temperature where the peak apex is located is the transition temperature ($T_{max}$) and the area under the peak provides the enthalpy of transition ($\Delta H$cal).

**TEM and 2D class averaging**. Electron microscopy was perform by NanoImaging Services (San Diego, CA) with a FEI Tecani T12 electron microscope, operated at 120 keV equipped with a FEI Eagle 4k × 4k CCD camera. SARS-CoV-2 S proteins were diluted to 2.5 μg mL$^{-1}$ in formulation buffer. The samples (3 μL) were applied to nitrocellulose-supported 400-mesh copper grids and stained with uranyl formate. Images of each grid were acquired at multiple scales to assess the overall distribution of the sample. High-magnification images were acquired at nominal magnifications of 110,000× (0.10 nm/pixel) and 67,000× (0.16 nm/pixel). The images were acquired at a nominal under focus of −1.4 μm to −0.8 μm (110,000×) and electron doses of ~25 e Å$^{-2}$.

For class averaging, particles were identified at high magnification prior to alignment and classification. The individual particles were selected, boxed out, and individual sub-images were combined into a stack to be processed using reference-free classification. Individual particles in the 67,000× high magnification images were selected using an automated picking protocol[17]. An initial round of alignments was performed for each sample, and from the alignment class averages that appeared to contain recognizable particles were selected for additional rounds of alignment. These averages were used to estimate the percentage of particles that

resembled single trimers and oligomers. A reference-free alignment strategy based on XMIPP processing package was used for particle alignment and classification[18].

**Kinetics of SARS-CoV-2 S binding to hACE2 receptor by BLI.** S-protein receptor binding kinetics were determined by BLI using an Octet QK384 system (Pall FortéBio, Fremont, CA). His-tagged hACE2 (2 µg mL$^{-1}$) was immobilized on nickel-charged Ni-NTA biosensor tips. After baseline, SARS-CoV-2 S-protein-containing samples were twofold serially diluted over a range of 4.7–300 nM and were allowed to associate for 600 s followed by dissociation for an additional 900 s. Data were analyzed with Octet software HT 101:1 global curve fit.

**Specificity of SARS-CoV-2 S binding to hACE2 receptor by ELISA.** Ninety-six-well plates were coated with 100 µL SARS-CoV-2 S protein (2 µg mL$^{-1}$) overnight at 4 °C. Plates were washed with phosphate-buffered saline with 0.05% Tween (PBS-T) buffer and blocked with TBS Startblock blocking buffer (Thermo Fisher Scientific). Histidine-tagged hACE2 and hDPP4 receptors were threefold serially diluted (5–0.0001 µg mL$^{-1}$) and added to coated wells for 2 h at room temperature. The plates were washed with PBS-T. Optimally diluted (1:4000) horseradish peroxidase (HRP) conjugated mouse anti-histidine was added and color developed by addition of and 3,3′,5,5′-tetramethylbenzidine (TMB) peroxidase substrate (T0440-IL, Sigma, St. Louis, MO, USA). Plates were read at an OD of 450 nm with a SpectraMax Plus plate reader (Molecular Devices, Sunnyvale, CA, USA) and data analyzed with SoftMax software. EC$_{50}$ values were calculated by 4-parameter fitting using GraphPad Prism 7.05 software (San Diego, CA, USA).

**Animal ethics statement.** The mouse immunogenicity studies were performed by Noble Life Sciences (Sykeville, MD). Noble Life Sciences is accredited by the Association for Assessment and Accreditation of Laboratory Animal Care (AAALACC International). The mouse SARS-CoV-2 challenge study was conducted at the University of Maryland BSL-3 containment facility (College Park, MD). The murine challenge study was conducted in accordance the University of Maryland Institutional Animal Care and Use Committee (IACUC) approved protocol. All animal procedures were in accordance with NRC Guide for the Care and Use of Laboratory Animals, the Animal Welfare Act, and the CDC/NIH Biosafety in Microbiological and Biomedical Laboratories. The olive baboon (*Papio cynocephalus Anubis*) study was performed at the University of Oklahoma Health Science Center (OUHSC). OUHSC is accredited by AAALACC International. Baboons were maintained and treated according to the Institutional Biosafety Committee guidelines. Baboon experiments were approved by the Institutional Animal Care and Use Committee (IACUC) and the Institutional Biosafety Committee of OUHSC. Studies were conducted in accordance with the National Institutes of Health Guide for Care and Use of Laboratory Animals (NIH publication 8023, Revised 1978).

**Animal husbandry.** Animal husbandry was conducted in accordance with each facilities SOPs. Mice were maintained in cages on racks equipped with an automatic watering system. Baboons were housed in individual stainless-steel cages with access to a watering system. Lighting was provided on a light/dark cycle approximately 12 h each day. The animal room temperature range was 73–74 °F and the relative humidity range was 36–49%. Animals were fed certified chow that had been analyzed for ensure no contaminants were in the feed. Water was available ad libitum throughout the studies.

**Mouse study designs.** Female BALB/c mice (7–9 weeks old, 17–22 grams, $n = 10$ per group) were immunized by intramuscular (IM) injection with a single dose (study day 14) or two doses spaced 14 days apart (study day 0 and 14) containing a dose range (0.01, 0.1, 1.0, or 10 µg) of NVX-CoV2373 with 5 µg saponin-based Matrix-M™ adjuvant (Novavax AB, Uppsala, SE). A separate group ($n = 10$) received two doses of 10 µg NVX-CoV2373 without adjuvant. A placebo group served as a non-immunized control. Serum was collected for analysis on study days −1, 13, 21, and 28. Vaccinated and control animals were intranasally challenged with live SARS-CoV-2 virus 42 days following one or two immunizations (study day 56).

To assess the T cell response mediated by Matrix-M adjuvant, groups of female BALB/c mice ($n = 6$ per group) were immunized IM with 10 µg NVX-CoV2373 with or without 5 µg Matrix-M adjuvant in two doses spaced 21 days apart. Spleens were collected 7 days after the second immunization (study day 28). A non-vaccinated group ($n = 3$) served as a control.

**Baboon study design.** Ten adult baboons (10–16 years of age) were randomized into four groups of 2–3/group and immunized by IM injection with 1, 5, or 25 µg NVX-CoV2373 with 50 µg Matrix-M adjuvant. A separate group was immunized with 25 µg NVX-CoV2373 without adjuvant. Animals were vaccinated with two doses spaced 21 days apart. Serum was collected on study days 0, 21, 28 and 35. For T cell analysis, peripheral blood mononuclear cells (PBMCs) were collected 7 days after the second immunization (study day 28). Subsequent to the start of the study, one animal tested positive for simian T-lymphotrophic virus and was therefore dropped from the study.

**SARS-CoV-2 challenge in mice.** The SARS-CoV-2 challenge portion of the study was performed at the University of Maryland BSL-3 containment facility (College Park, MD). BALB/c mice were made permissive to infection with SARS-CoV-2 by transduction with human Ad5/hACE2 receptor[21,22]. Mice were transduced intranasally with $2.5 \times 10^8$ pfu Ad/CMVhACE2 (VVC-McCray-7580, University of Iowa Vector Core) 38 days after immunization. At 4 days post transduction, mice were anesthetized by intraperitoneal injection with 50 µL of a mix of xylazine (0.38 mg/mouse) and ketamine (1.3 mg/mouse) diluted in phosphate buffered saline (PBS). Mice were intranasally inoculated with $1.5 \times 10^5$ pfu of SARS-CoV-2 in 50 µL divided between nares. Challenged mice were weighed on day of infection and daily for up to 7 dpi. At days 4 and 7 dpi, five mice were sacrificed from each vaccination and control group, and lungs were harvested to determine viral titer by a plaque assay and prepared for histological scoring.

**SARS-CoV-2 plaque assay.** SARS-CoV-2 lung titers were quantified by homogenizing harvested lungs in PBS (Quality Biological Inc.) using 1.0 mm glass beads (Sigma Aldrich) and a Beadruptor (Omini International Inc.). Homogenates were added to near-confluent Vero E6 cultures and SARS-CoV-2 virus titers determined by counting plaque forming units using a 6-point dilution curve.

**Anti-SARS-CoV-2 S IgG by ELISA.** An ELISA was used to determine anti-SARS-CoV-2 S IgG titers. Briefly, 96-well microtiter plates (Thermo Fisher Scientific, Rochester, NY, USA) were coated with 1.0 µg mL$^{-1}$ of SARS-CoV-2 S protein. Plates were washed with PBS-T and blocked with TBS Startblock blocking buffer (Thermo Fisher Scientific). Mouse, baboon, or human serum samples were serially diluted ($10^{-2}$ to $10^{-8}$) and added to the blocked plates before incubation at room temperature for 2 h. Following incubation, plates were washed with PBS-T and HRP-conjugated goat anti-mouse IgG (1:5000) or goat anti-human IgG (1:2000) (Southern Biotech, Birmingham, AL, USA) added for 1 h. Plates were washed with PBS-T and TMB peroxidase substrate (T0440-IL, Sigma, St. Louis, MO, USA) was added. Reactions were stopped with TMB stop solution (ScyTek Laboratories, Inc. Logan, UT). Plates were read at OD 450 nm with a SpectraMax Plus plate reader (Molecular Devices, Sunnyvale, CA, USA) and data analyzed with SoftMax software. EC$_{50}$ values were calculated by 4-parameter fitting using SoftMax Pro 6.5.1 GxP software. Individual animal anti-SARS-CoV-2 S IgG titers and group GMTs and 95% confidence intervals (±95% CI) were plotted using GraphPad Prism 7.05 software.

**ACE2-receptor-blocking antibodies.** ACE2-receptor-blocking antibodies were determined by ELISA. Ninety-six-well plates were coated with 1.0 µg mL$^{-1}$ SARS-CoV-2 S protein overnight at 4 °C. Serially diluted serum from groups of immunized mice, baboons, or humans were added to coated wells and incubated for 2 h at room temperature. After washing, 30 ng mL$^{-1}$ of histidine-tagged hACE2 or hDPP4 was added to wells for 1 h at room temperature. HRP-conjugated mouse anti-histidine IgG (1:4000) (Southern Biotech, Birmingham, AL, catalog no. 4603-05) was added followed by washing prior to addition of TMB substrate. Plates were read at OD 450 nm with a SpectraMax plus plate reader (Molecular Devices, Sunnyvale, CA, USA) and data analyzed with SoftMax software. Serum dilution resulting in a 50% inhibition of receptor binding was used to calculate titer determined using 4-parameter fitting with GraphPad Prism 7.05 software.

**SARS-CoV-2 neutralization assay.** SARS-CoV-2 was handled in a select agent ABSL3 facility at the University of Maryland, School of Medicine. Mouse or baboon sera were diluted 1:20 in Vero E6 cell growth media and further diluted 1:2 to 1:40960. SARS-CoV-2 (MOI of 0.01 pfu per cell) was added and the mixture incubated for 60 min at 37 °C. Vero E6 media was used as negative control. The inhibitory capacity of each serum dilution was assessed for CPE. The endpoint titer was reported as the dilution at which CPE first became visible at 3 dpi.

**ELISpot assay.** Murine IFN-γ and IL-5 ELISpot assays were performed following manufacturer's procedures for mouse IFN-γ and IL-5 ELISpot kits (3321-2H and 3321-2A, Mabtech, Cincinnati, OH). Briefly, $10^5$ splenocytes in a volume of 200 µL were stimulated with NVX-CoV2373 protein or peptide pools (PP) of 15-mer peptides with 11 overlapping amino acids covering the entire spike protein sequence (JPT, Berlin, Germany) in plates that were pre-coated with anti-IFN-γ (15 µg mL$^{-1}$) or anti-IL-5 (15 µg mL$^{-1}$) antibodies. Detection second antibody clone RS-6A2 IFN-γ (1 µg mL$^{-1}$) and clone TRFK4 (1 µg mL$^{-1}$). Each stimulation condition was carried out in triplicate. Assay plates were incubated overnight at 37 °C in a 5% CO$_2$ incubator and developed using BD ELISpot AEC substrate set (BD Biosciences, San Diego, CA). Spots were counted and analyzed using an ELISpot reader and ImmunoSpot software v5.03 (Cellular Technology, Ltd., Shaker Heights, OH). The number of IFN-γ- or IL-5-secreting cells was obtained by subtracting the background number in the medium controls. Data shown in the graph are the average of triplicate wells.

Similarly, baboon IFN-γ and IL-4 assays were carried out using NHP IFN-γ and Human IL-4 assay kits from Mabtech. For IFN-γ, coating antibody human IFN-γ 3420-2H (15 µg mL$^{-1}$) and detection antibody clone 7-B6-1 (1 µg mL$^{-1}$). For IL-4, coating antibody human IL-43410-2H (clone IL4-I (15 µg mL$^{-1}$) and detection

antibody clone IL4-II (1 μg mL$^{-1}$)). PBMCs were collected on day 7 following the second immunization (day 28). Assays were performed in triplicate wells.

**Surface and intracellular cytokine staining**. For surface staining, murine splenocytes were first incubated with an anti-CD16/32 antibody to block the Fc receptor. To characterize Tfh cells, splenocytes were incubated with the following antibodies or dye: BV650-conjugated anti-CD3 (1:25), APC-H7-conjugated anti-CD4 (1:25), FITC-conjugated anti-CD8 (1:50), Percp-cy5.5-conjugated anti-CXCR5 (1:10), APC-conjugated anti-PD-1 (1:50), Alexa Fluor 700-conjugated anti-CD19 (1:25), PE-conjugated anti-CD49b (1:50) (BD Biosciences, San Jose, CA), and the yellow LIVE/DEAD® dye (1:300) (Life Technologies, NY). To stain GC B cells, splenocytes were labeled with FITC-conjugated anti-CD3 (1:25), PerCP-Cy5.5-conjugated anti-B220 (1:50), APC-conjugated anti-CD19 (1:25), PE-cy7-conjugated anti-CD95 (1:00), and BV421-conjugated anti-GL7 (1:50) (BD Biosciences) and the yellow LIVE/DEAD® dye (1:300) (Life Technologies, NY).

For ICCS of murine splenocytes, cells were cultured in a 96-well U-bottom plate at $2 \times 10^6$ cells per well. The cells were stimulated with NVX-CoV2373 or pools of a 15-mer PP as described above (JPT, Berlin, Germany). The plate was incubated 6 h at 37 °C in the presence of BD GolgiPlug™ and BD GolgiStop™ (BD Biosciences). Cells were labeled with murine antibodies against BV650 CD3 (1:25), APC-H7 CD4 (1:25), FITC CD8 (1:50), Alexa Fluor 700 CD44 (1:25), and PE CD62L (1:50) (BD Pharmingen, CA) and the yellow LIVE/DEAD® dye (1:300). After fixation with Cytofix/Cytoperm (BD Biosciences), cells were incubated with PerCP-Cy5.5-conjugated anti-IFN-γ (1:25), BV421-conjugated anti-IL-2 (1:50), PE-cy7-conjugated anti-TNF-α (1:25), and APC-conjugated anti-IL-4 (1:25) (BD Biosciences). All stained samples were acquired using a LSR-Fortessa flow cytometer (Becton Dickinson, San Jose, CA) and the data were analyzed with FlowJo software version Xv10 (Tree Star Inc., Ashland, OR) (Supplementary Fig. 3).

For ICCS, baboon PBMCs were collected 7 days after the second immunization (day 28) and stimulated as described above with NVX-CoV2373. Cells were labeled with human/NHP antibodies BV650-conjugated anti-CD3 (1:10), APC-H7-conjugated anti-CD4 (1:10), APC-conjugated anti-CD8 (1:10), BV421-conjugated anti-IL-2 (1:25), PerCP-Cy5.5-conjugated anti-IFN-γ (1:10), PE-cy7-conjugated anti-TNF-α (1:50) (BD Biosciences), and the yellow LIVE/DEAD® dye (1:300).

**Histopathology**. Mice were euthanized at 4 and 7 days following SARS-CoV-2 challenge. The lungs were fixed with 10% formalin, and sections were stained with hematoxylin and eosin (H&E) for histological examination. Slides were examined in a blinded fashion. Sections were scored on a severity scale of 0–5 for total, perivascular, and peribronchiolar inflammation.

**COVID-19 convalescent serum**. Convalescent serum samples were provided by Dr. Pedro A. Piedra (Baylor College of Medicine, Houston, TX, USA). The serum samples were provided to Novavax, Inc. through a Master Trial Agreement, and all participants in this serology study provided written informed consent, through approval from the Baylor College of Medicine Institutional Review Board (IRB). Samples (n = 32) were collected from COVID-19 patients 18–79 years of age 4–6 weeks after testing positive for SARS CoV-2 by the anti-S IgG ELISA and virus-neutralizing antibody assay (99% CPE). Symptoms ranged from asymptomatic, mild to moderate symptoms, to severe symptoms requiring hospitalization[30]. Sera were analyzed for anti-SARS-CoV-2 S IgG and hACE2-receptor-inhibiting antibody levels.

**Statistical analysis**. Statistical analyses were performed with GraphPad Prism 7.05 software. Serum antibody titers were plotted for individual animals and the GMT and 95% CI or the means ± SD as indicated in the figure. Student's $t$-test was used to determine differences between paired groups. Weight change between immunized and placebo groups was determined for each day using a Student's $t$-test. $p$-Values ≤0.05 were considered as statistically significant.

**Reporting summary**. Further information on research design is available in the Nature Research Reporting Summary linked to this article.

## Data availability
All data generated or analyzed during this study are included in this published article and the supplementary information files. All other relevant data are available from the corresponding author on reasonable request. Source data are provided with this paper.

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

## Acknowledgements

Funding for certain studies was provided by the Coalition for Epidemic Preparedness Innovations, PO Box 123, Torshov, 0412 Oslo, Norway.

## Author contributions

G.S., G.G., J.H.T., N.P., R.H., H.Z., M.G.X., A.D.P., M.J.M., M.B.F., and L.E. contributed to conceptualization of experiments, generation of data and analysis, and interpretation of the results. J.H.T., R.H., N.P., S.W., H.H., J.L., J.N., B.Z., K.J., S.M., R.K., M.W., W.M., S.K.S., B.E., S.B., C.J.W., and H.Z. performed the experiments. A.D.P., M.G.X., and J.P. coordinated the projects. M.B.F., A.D.P., M.J.M., L.F., P.A.P., K.L.B., L.S., G.G., G.S., and L.E. contributed to drafting and making critical revisions with the help of others.

## Competing interests

Authors J.H.T., N.P., H.Z., A.D.P., J.N., M.G.X., B.Z., K.J., S.M., R.K., M.W., W.M., S.K.S., B.E., M.J.M., S.B., C.J.W., L.F., K.L.B., L.S., G.G., L.E., and G.S. are current or past employees of Novavax, Inc, a for-profit organization, and these authors own stock or hold stock options. M.B.F., R.H., S.W., J.L., H.H., P.A.P., and J.P. declare no competing interests.
