## [Peer Review File · Nature Communications]

Reviewers' Comments:

Reviewer #1:

Remarks to the Author:

In their manuscript, Tian and colleagues describe the process and result of the development of a full-length S protein-based subunit vaccine. After the evaluation of three subunit vaccine candidates for SARS-CoV-2 through in vitro study, proline-stabilized prefusion NVX-CoV2373 was chosen, in combination with Matrix-M adjuvant, to be further evaluated in animal models. The vaccination induced robust neutralizing antibodies and a balanced T cell response, also protected ACE2-transduced mice from SARS-CoV-2 infection. The quality of the data is good and the manuscript is well written. This is the same vaccine candidate that is currently being assessed in clinical trials by Novavax, also the first full-length S based subunit vaccine, as far as I know, that went to clinical trial. These data were produced in the remarkably rapid development over only several months. Clearly this work is of importance and urgency. There are two questions.

1. Based on the SDS-PAGE gel, the WT protein produced in insect cell was not cleaved, although the furin site was maintained. Can the authors explain what causes the difference between WT and BV2365, for example, in stability?
2. The 'PP' version NVX-CoV2373 is in prefusion conformation under TEM. Has the author tested the WT and BV2365? Expression and purification of full-length S protein is more challenging than ectodomain. It'll be better if the author can describe the expression level and protein conformation for each sample.

Reviewer #2:

Remarks to the Author:

In the manuscript titled "SARS-CoV-2 spike glycoprotein vaccine candidate NVX-CoV2373 elicits 2 immunogenicity in baboons and protection in mice" Tian JH, Patel N, and Haupt R et al., examine stability, immunogenicity, and protective efficacy of Matrix-M adjuvanted SARS-coV-2 spike sub-unit vaccine in mouse and baboon models. The authors generate a double mutant SARS-2 spike protein (NVX-CoV2373) with mutation at furin site and 2Proline residues to obtain a stable pre-fusion spike protein. The authors test the immunogen for its structure and stability and find that NVX-CoV2373 is more stable than furin site-only mutant protein BV2365 when subjected to different stress conditions. While immunization of mice with a single dose adjuvanted protein elicited very low level of neutralizing antibodies, a two dose immunization induced high titers of neutralizing antibodies, particularly in adjuvanted group. Further, two dose immunization with 1µg or high antigen dose provided (sterilizing?) protective immunity, generated spike protein-specific CD4 and CD8 T cell responses without vaccine enhanced disease. The authors also test immunization in baboons. Matrix-M adjuvant Spike subunit antigen induced strong antibody and T cell response. Collectively, this is a largely well designed study, with two dose immunization strategy generating robust protective immunity against SARS-coV-2 in animals. There are a few weaknesses that need to be addressed.

Major Comments:

- 1) Please provide the rationale for not including WT spike protein or BV2365 as candidate/ss to demonstrate the superior efficacy of NVX-CoV2373.
- 2) Looking at the antibody and T cell responses, the adjuvant has a dramatic effect on the generation of these responses. However, an adjuvant only group is not included in the study. Although it is unlikely that adjuvant will induce specific immunity, this control groups is required for these studies.
- 3) Line 194/Figure 4: Looks like antibody responses were measured at 21 or 28 day post-immunization but the mouse challenge was not done until 52 days post-immunization. Please provide rationale for these different strategies. What was neutralizing antibody and specific T cell response at this time point?
- 4) Virus titration at day 2 infection would be helpful to assess if the vaccination induces sterilizing immunity. This reviewer suggests authors to include day 2 titers if available.

- 5) Figure 8: It is surprising to see no neutralizing antibody induction in NVX-CoV2373-only two dose groups. Similar to mouse studies, adjuvant-only control is not included. Please address these issues.
6) It is essential to include virus titers or genomic RNA levels in control and immunized baboons.

Reviewer #3:

Remarks to the Author:

SARS-CoV-2 spike glycoprotein vaccine candidate NVX-CoV2373 elicits immunogenicity in baboons and protection in mice.

Tian et al.

In this manuscript the authors present the characterization of two SARS-CoV-2 spike proteins, one that has been engineered with mutation that abrogate the furin cleavage site (BV2365) and the other (NVX-2373) with additional mutation that include two consecutive prolines to stabilize the protein in the prefusion conformation. The first part of the manuscript is devoted to the biophysical characterization of the two proteins as well as showing their affinity characteristics for binding to the human ACE2 protein. In the second part the immunological characterization of NVX-2373 as the antigen for a vaccine in the presence or absence of adjuvant (Matrix-M) using a mouse (BALB/c) model and a baboon model. Protection and vaccine associated enhanced respiratory disease were evaluated in a SARS-CoV-2 mouse model based on the transduction of hACE2 via intranasal infection with a recombinant adenovirus.

The paper is important since this is a different platform among many in the race for a coronavirus vaccine. The spike protein is made in baculovirus system, a well-known platform for scale up vaccine production. It is clear that this requires the addition of adjuvant to achieve a measurable immunological response. The results obtained showed a moderate difference between BV2365 and NVX-2373 in terms of the stability of the protein under different stress conditions (measured as binding to hACE2). Thus, Figure 3 define the antigen used in the following vaccination experiments. However, with the data provided, this decision seems premature, specially from the point of view that the antigenicity of NVX-2373 is very low in the absence of adjuvant. Demonstration of immunological superiority of NVX-2373 vs BV2365 in addition would be a more powerful result for making that decision.

The weakest part of the manuscript is the experimental approach followed to test the efficacy and safety of the vaccine. The murine model seems not well established, no publications of this model are cited, and there is no demonstration that the hACE2 protein is equally and homogeneously expressed in the respiratory tract of the animals at the time of SARS challenge. In fact, as recognized by the authors in the discussion, alteration of lung pathology by Ad infection prior to SARS-CoV2 challenge could alter many of the inflammatory pathways in the lung, enhancing or masking the response to SARS challenge. A second model of efficacy and VAERD, or a more detail characterization of SARS-CoV-2 infection in this model (including ACE2-transduce, uninfected controls for pathology) would greatly improve the manuscript.

Other points

Please explain why there are different protocols of vaccination, bleeding, challenge and sacrifice. A scheme that reflect the timeline of each study would be handy for the reader.

Please explain why there is no correlation between NA titers, protection, and % weight change (e.g., 1 dose, 0.1 µg + adjuvant, where there are none NA; two doses 0.1 and 1 µg, where maximum NA are not effective in fully reducing weight change).

Quality of pathology depicted in Figure 7 is marginal. No depiction of differences between prime and prime boost regimen. On day 7 in the prime-boost regimen it seems like there is more bronchial infiltrates in vaccinated animals that in the placebo control; size of the vasculature compared should

be similar, and also increase the area of the parenchyma depicted (alveoli) for adequate comparison of % of consolidated inflammation. Inclusion of graph with the mean pathology scores from each group will support statement in line 229. Control showing lung histology in transduced, unchallenged animal of the corresponding days will help for comparison. Pathology figure should define the magnification.

Statement on discussion, line 23 "Low, suboptimal..." seems not supported by data (no description in the dose of vaccine used in Figure 5).

The following is a point-by-point response to Reviewers:

Reviewer #1: *In their manuscript, Tian and colleagues describe the process and result of the development of a full-length S protein-based subunit vaccine. After the evaluation of three subunit vaccine candidates for SARS-CoV-2 through in vitro study, proline-stabilized prefusion NVX-CoV2373 was chosen, in combination with Matrix-M adjuvant, to be further evaluated in animal models. The vaccination induced robust neutralizing antibodies and a balanced T cell response, also protected ACE2-transduced mice from SARS-CoV-2 infection. The quality of the data is good and the manuscript is well written. This is the same vaccine candidate that is currently being assessed in clinical trials by Novavax, also the first full-length S based subunit vaccine, as far as I know, that went to clinical trial. These data were produced in the remarkably rapid development over only several months. Clearly this work is of importance and urgency.*

There are two questions.

1. *Based on the SDS-PAGE gel, the WT protein produced in insect cell was not cleaved, although the furin site was maintained. Can the authors explain what causes the difference between WT and BV2365, for example, in stability?*

Response: In Sf9 cells, type 1 fusion glycoproteins with furin cleavage sites are known to be partially or not fully cleaved likely due to differences in mammalian and insect furin proteases. With respect to stability of wild type (WT) and BV2365, the SARS-CoV-2 spike (S) is known to be metastable and mutations are required to prevent unfolding into a post-fusion conformation (Wrapp, D et al. Science. 2020, 367; 1260-1263).

2. *The 'PP' version NVX-CoV2373 is in prefusion conformation under TEM. Has the author tested the WT and BV2365? Expression and purification of full-length S protein is more challenging than ectodomain. It'll be better if the author can describe the expression level and protein conformation for each sample.*

Response: We thank the reviewer for their suggestion, however the instability of WT SARS-CoV-2 S is well established and was not the focus of this study. Our study was focused on identifying a full-length S vaccine candidate NVX-CoV2373 that was thermostable, formed nanoparticles, and in a prefusion trimeric conformation that compared well to the structure of truncated S trimers (Wrapp, D et al. Science. 2020, 367; 1260-1263). In our expression and purification studies, the wild type version was unstable and often cleaved while the pre-fusion stabilized version was stable across many heat and purification parameters. For our analysis, and because the goal is to mass produce this vaccine to hundreds of millions of doses, the stability of the particle is critical.

Reviewer #2: *In the manuscript titled "SARS-CoV-2 spike glycoprotein vaccine candidate NVX-CoV2373 elicits 2 immunogenicity in baboons and protection in mice" Tian JH, Patel N, and Haupt R et al., examine stability, immunogenicity, and protective efficacy of Matrix-M adjuvanted SARS-coV-2 spike sub-unit vaccine in mouse and*

baboon models. The authors generate a double mutant SARS-2 spike protein (NVX-CoV2373) with mutation at furin site and 2Proline residues to obtain a stable pre-fusion spike protein. The authors test the immunogen for its structure and stability and find that NVX-CoV2373 is more stable than furin site-only mutant protein BV2365 when subjected to different stress conditions. While immunization of mice with a single dose adjuvanted protein elicited very low level of neutralizing antibodies, a two dose immunization induced high titers of neutralizing antibodies, particularly in adjuvanted group. Further, two dose immunization with 1µg or high antigen dose provided (sterilizing?) protective immunity, generated spike protein-specific CD4 and CD8 T cell responses without vaccine enhanced disease. The authors also test immunization in baboons. Matrix-M adjuvant Spike subunit antigen induced strong antibody and T cell response. Collectively, this is a largely well designed study, with two dose immunization strategy generating robust protective immunity against SARS-coV-2 in animals. There are a few weaknesses that need to be addressed.

Major comments:

1. *Please provide the rationale for not including WT spike protein or BV2365 as candidate/ss to demonstrate the superior efficacy of NVX-CoV2373.*

Response: The WT spike protein is very unstable, unfolding into a post-fusion conformation rapidly, and the BV2365 without the 2P mutation is unstable. In our expression and purification studies, the wild type version was unstable and often cleaved while the pre-fusion stabilized version was stable across many heat and purification parameters. For our analysis, and because the goal is to mass produce this vaccine to hundreds of millions of doses, the stability of the particle is critical. Thus, neither construct would be a viable COVID-19 vaccine candidate, thus not included in a mouse challenge study requiring a BSL3 facility.

2. *Looking at the antibody and T cell responses, the adjuvant has a dramatic effect on the generation of these responses. However, an adjuvant only group is not included in the study. Although it is unlikely that adjuvant will induce specific immunity, this control groups is required for these studies.*

Response: The mechanism of action of saponin-based Matrix-M adjuvant has been reported and includes recruitment of leukocytes to local draining lymph nodes and spleen, and the induction of local transient response with recruitment, activation, and maturation of monocytes, dendritic cells, T cells, B cells, and natural killer cells (Reimer, J. M. et al. PLoS One. 2012, 7; e41451; Magnusson, S.E. et al. Immunol. Res. 2018, 66; 224-233). While we agree that an adjuvant-only arm could have been included as another control, Matrix-M alone has been used in a variety of animal and human vaccine studies and does not induce antigen-specific antibody or T-cell responses (Bengtsson, KL et al Vaccine. 2016, 34; 1927-1935); a brief discussion of the adjuvant mode of action and references were added to the Discussion section of the manuscript.

3. *Line 194/Figure 4: Looks like antibody responses were measured at 21 or 28 day post-immunization but the mouse challenge was not done until 52 days post-immunization. Please provide rationale for these different strategies. What was neutralizing antibody and specific T cell response at this time point?*

Response: The primary purpose of this study was to assess the immunogenicity of NVX-CoV2373 vaccine using a prime and prime/boost regimen as well as protection against SARS-CoV-2 challenge. The interval between the immunization and challenge was dictated by the logistics of transfer of vaccinated animals from the CRO, Noble Life Sciences, to the University of Maryland BSL-3 containment facility. In addition, animals were required to be acclimated prior to transduction with Ad/hACE2 and subsequent challenge.

4. *Virus titration at day 2 infection would be helpful to assess if the vaccination induces sterilizing immunity. This reviewer suggests authors to include day 2 titers if available.*

Response: We agree that the data are insufficient to claim sterilizing immunity and have removed this statement from the paper. Unfortunately, sufficient quantities of vaccinated mice were not available for this experiment to take multiple time points.

5. *Figure 8: It is surprising to see no neutralizing antibody induction in NVX-CoV2373-only two dose groups. Similar to mouse studies, adjuvant-only control is not included. Please address these issues.*

Response: In non-adjuvanted study arms, the two baboons had anti-S and neutralizing antibodies above the LOD (Figure 8B and 8D) and in mice, all of the animals immunized with NVX-CoV2373 only developed anti-S IgG, and pooled serum exhibited >1000 neutralizing antibody titers after two doses (Figure 4D). As mentioned in point 1 above, based on our prior experience, an adjuvant-only control would not be expected to induce antigen-specific responses or protection.

6. *It is essential to include virus titers or genomic RNA levels in control and immunized baboons.*

Response: The baboons in this study were not challenged with live SARS-CoV-2, only vaccinated. They were assessed for antibody and T-cell responses (Figure 8). Virus titers or genomic RNA levels in the baboons are not applicable to this study.

Reviewer #3: *In this manuscript the authors present the characterization of two SARS-CoV-2 spike proteins, one that has been engineered with mutation that abrogate the furin cleavage site (BV2365) and the other (NVX-2373) with additional mutation that include two consecutive prolines to stabilize the protein in the prefusion conformation. The first part of the manuscript is devoted to the biophysical characterization of the two proteins as well as showing their affinity characteristics for binding to the human ACE2 protein. In the second part the immunological characterization of NVX-2373 as the*

antigen for a vaccine in the presence or absence of adjuvant (Matrix-M) using a mouse (BALB/c) model and a baboon model. Protection and vaccine associated enhanced respiratory disease were evaluated in a SARS-CoV-2 mouse model based on the transduction of hACE2 via intranasal infection with a recombinant adenovirus.

The paper is important since this is a different platform among many in the race for a coronavirus vaccine. The spike protein is made in baculovirus system, a well-known platform for scale up vaccine production. It is clear that this requires the addition of adjuvant to achieve a measurable immunological response. The results obtained showed a moderate difference between BV2365 and NVX-2373 in terms of the stability of the protein under different stress conditions (measured as binding to hACE2). Thus, Figure 3 define the antigen used in the following vaccination experiments. However, with the data provided, this decision seems premature, specially from the point of view that the antigenicity of NVX-2373 is very low in the absence of adjuvant. Demonstration of immunological superiority of NVX-2373 vs BV2365 in addition would be a more powerful result for making that decision.

- 1. The weakest part of the manuscript is the experimental approach followed to test the efficacy and safety of the vaccine. The murine model seems not well established, no publications of this model are cited, and there is no demonstration that the hACE2 protein is equally and homogeneously expressed in the respiratory tract of the animals at the time of SARS challenge. In fact, as recognized by the authors in the discussion, alteration of lung pathology by Ad infection prior to SARS-CoV2 challenge could alter many of the inflammatory pathways in the lung, enhancing or masking the response to SARS challenge. A second model of efficacy and VAERD, or a more detail characterization of SARS-CoV-2 infection in this model (including ACE2-transduce, uninfected controls for pathology) would greatly improve the manuscript.*

Response: As SARS-CoV-2 emerged, a rapid mouse model was generated based on previous work in MERS-CoV, another pathogenic coronavirus that does not naturally replicate or cause disease in mice. Murine MERS-CoV models utilized an Adenovirus expressing hDPP4 to render mice permissive to infection. Based on that model, the same group led by Dr. Stan Perlman generated the Ad/hACE2 model. At the time of our initial manuscript submission, their paper was not published; however, studies utilizing this model have now been published in *Cell* (Hassan, et al. *Cell*. 2020, 182; 744-753) and *Nature* (Zost, et al. *Nature*. 2020, 584; 443-449). As stated in the *Cell* paper from Dr. James Crowe's lab, this a robust SARS-CoV-2 challenge model. We believe this is a suitable challenge model to evaluate a COVID-19 vaccine candidate. Additional references and a statement regarding limitations of the Ad/ACE2 model were added to the Discussion section of the manuscript. Additionally, pathology images for Ad/hACE2 transduced but uninfected mouse lungs of age- and sex-matched BALB/c mice from a different experiment have been added to Figure 5.

Other points:

- 2. Please explain why there are different protocols of vaccination, bleeding, challenge and sacrifice. A scheme that reflect the timeline of each study would be handy for the reader.*

Response: Study design schemes are described in the Methods and we have also added study timelines in Figures 4A, 6A, and 8A. As for different protocols, for example T-cell responses were measured day 7 after a prime and boost immunization when cellular responses were more optimal. We don't believe challenge and T-cell studies are required to be on the same schedule. We hope this clarifies the methods and timing used in the manuscript.

- 3. Please explain why there is no correlation between NA titers, protection, and % weight change (e.g., 1 dose, 0.1 μg + adjuvant, where there are none NA; two doses 0.1 and 1 μg , where maximum NA are not effective in fully reducing weight change).*

Response: We thank the reviewer for their observation and it is something we have been contemplating about the model. Correlates of protection are important and we do not claim to have identified any specific correlates in this study. Protection is likely complex when weight loss, neutralizing antibody, virus titer and T-cell responses all play a role in disease. We do show a strong correlation between neutralizing antibodies and anti-S IgG titers (Figure 8E) which is evidence that the vaccine is inducing functional antibody responses. It is high priority to search for correlates of protection in recently initiated human efficacy studies with NVX-CoV2373 vaccine.

- 4. Quality of pathology depicted in Figure 7 is marginal. No depiction of differences between prime and prime boost regimen. On day 7 in the prime-boost regimen it seems like there is more bronchial infiltrates in vaccinated animals that in the placebo control; size of the vasculature compared should be similar, and also increase the area of the parenchyma depicted (alveoli) for adequate comparison of % of consolidated inflammation. Inclusion of graph with the mean pathology scores from each group will support statement in line 229.*

Response: We generally agree and added a discussion of pulmonary inflammation as one of the limitations of the Ad/ACE2 model as described by Hassan, AO et al Cell. 2020, 182; 744-753 and Zost, SJ et al. Nature. 2020, 584; 443-449. As requested, the mean histopathology scores from each vaccination and placebo group have been added and is described in Supplementary Figure 1.

- 5. Control showing lung histology in transduced, unchallenged animal of the corresponding days will help for comparison. Pathology figure should define the magnification.*

Response: Transduced unchallenged lungs have been added to Figure 5 as recommended and the magnification of histology images has been included in Figure 5 legend. We hope this clarifies the interpretation of the pathology.

6. *Statement on discussion, line 23 “Low, suboptimal...” seems not supported by data (no description in the dose of vaccine used in Figure 5).*

Response: We agree and revised the statement in the Discussion section to read “There was no evidence of VAERD in challenged mice immunized with NVX-CoV2373.”

Reviewers' Comments:

Reviewer #2:

Remarks to the Author:

The authors provided explanation for this reviewers comments. While there is no additional data included, the response is acceptable.

Authors mention that they did not use WT or BV2365 spike protein constructs because they are unstable and wouldn't be a viable COVID-19 vaccine candidate. This is perhaps a biased opinion considering NVX-CoV2373-alone induces very low levels of neutralizing antibody responses, and that the success of this vaccine candidate largely depends on adjuvant used in the study. Additionally, several other vaccine candidates being tested use WT-spike as the immunogen that show good specific antibody and T cell response.

Overall this is a good study that shows NVX-CoV2373+adj protective efficacy in animal models.

Reviewer #3:

Remarks to the Author:

The authors have answered to previous request/concerns.

Remaining point:

In the abstract, results in mice and baboons should be described in two separate sentences(lanes 48 to 54) and in the mentioned order.

A graph showing IL-4 ELISPOT using baboons cells should be incorporated to cover the statement "nonhuman primate and mice models induces a Th1 dominant B- and T-cell response" (lane 93).

The following are point-by-point responses to reviewer comments and suggestions (NCOMMS-20-26258B).

Reviewer #2 (Remarks to the Author):

The authors provided explanation for this reviewers comments. While there is no additional data included, the response is acceptable.

Authors mention that they did not use WT or BV2365 spike protein constructs because they are unstable and wouldn't be a viable COVID-19 vaccine candidate. This is perhaps a biased opinion considering NVX-CoV2373-alone induces very low levels of neutralizing antibody responses, and that the success of this vaccine candidate largely depends on adjuvant used in the study. Additionally, several other vaccine candidates being tested use WT-spike as the immunogen that show good specific antibody and T cell response.

Overall this is a good study that shows NVX-CoV2373+adj protective efficacy in animal models.

Response: The authors appreciate the reviewers' comments. No changes to the manuscript are required.

Reviewer #3 (Remarks to the Author):

The authors have answered to previous request/concerns.

Remaining point:

In the abstract, results in mice and baboons should be described in two separate sentences(lanes 48 to 54) and in the mentioned order.

Response: The abstract was modified to separate the description of the mouse and baboon results. As requested, the following sentence was added to the abstract "In baboons, low-dose levels of NVX-CoV2373 with Matrix-M was also highly immunogenic and elicited high titer anti-S antibodies and functional antibodies that block S-protein binding to hACE2 and neutralize virus infection and antigen-specific T cells."

A graph showing IL-4 ELISPOT using baboons cells should be incorporated to cover the statement "nonhuman primate and mice models induces a Th1 dominant B- and T-cell response" (lane 93).

Response: The sentence was modified and the term “Th1 dominant” was deleted. The revised sentence reads as follows: “Here we show that administering NVX-CoV2373 with Matrix-M adjuvant in a nonhuman primate and mice models induces a B- and T-cell responses, hACE2 receptor blocking antibodies, and SARS-CoV-2 neutralizing antibodies.”